



# Isoneutral control of effective diapycnal mixing in numerical ocean models with neutral rotated diffusion tensors

Antoine Hochet[1], Rémi Tailleux[1], David Ferreira[1], and Till Kuhlbrodt[1]

[1]University of Reading

*Correspondence to:* a.hochet@reading.ac.uk

**Abstract.** The current view about the mixing of heat and salt in the ocean is that it should be parameterised by means of a rotated diffusion tensor based on mixing directions parallel and perpendicular to the local neutral vector. However, the impossibility to construct a density variable in the ocean that is exactly neutral because of the coupling between thermobaricity and density-compensated temperature/salinity anomalies implies that the effective diapycnal diffusivity experienced by any possible density variable is partly controlled by isoneutral diffusion when using neutral rotated diffusion. Here, this effect
is quantified by evaluating the effective diapycnal diffusion coefficient for five widely used density variables: Jackett and McDougall (1997) $\gamma^n$, Lorenz reference state density $\rho_{ref}$ of Winters and D'Asaro (1996); Saenz et al. (2015), and three potential density variables $\sigma_0$, $\sigma_2$ and $\sigma_4$.Computations use the World Ocean Circulation Experiment climatology, assuming either a uniform value for isoneutral mixing or spatially varying values inferred from an inverse calculation. Isopycnal mixing
contributions to the effective diapycnal mixing yields values systematically larger than $10^{-3}$ m$^2$/s in the deep ocean for all density variables, with $\gamma^n$ suffering the least from the isoneutral control of effective diapycnal mixing, and $\sigma_0$ the most. These high values are due to spatially localised large values of non-neutrality, mostly in the deep Southern Ocean. Removing only 5% of these high values on each density surface reduces the effective diapycnal diffusivities to less than $10^{-4}$ m$^2$/s. This work highlights the potential pitfalls of estimating diapycnal diffusivities by means of Walin-like water masses analysis or in using
Lorenz reference state for diagnosing spurious numerical diapycnal mixing.

## 1 Introduction

Simulations of climate change by means of coupled ocean-atmosphere numerical models are sensitive to parameterisations of oceanic sub-grid scale mixing of heat and salt. Indeed, subgridscale mixing processes directly control ocean heat uptake, the strength of the Atlantic meridional overturning circulation, and the poleward heat transport e.g., Kuhlbrodt and Gregory
(2012). Historically, early numerical ocean models had used a diffusion tensor based on mixing heat and salt with different mixing diffusivities in the horizontal and vertical directions. Following Veronis (1975), it has been generally assumed that such an approach causes spurious upwelling in western boundary currents owing to the unphysical diapycnal mixing component due to the large horizontal mixing across sloping isopycnal surfaces, the so-called "Veronis effect". Indeed, the diffusive flux of any mathematically well-defined material density variable $\gamma(S, \theta)$, where $\theta$ is the potential temperature and $S$ the (practical)



salinity, for such a mixing tensor is given by:

$$\mathbf{F}_\gamma = -K_H \left[ \nabla\gamma - (\nabla\gamma \cdot \mathbf{k})\mathbf{k} \right] - K_V (\nabla\gamma \cdot \mathbf{k})\mathbf{k}, \tag{1}$$

where $K_H$ and $K_V$ are the horizontal and vertical mixing coefficients respectively, and $\mathbf{k}$ the unit normal vector pointing upwards. Therefore, the diapycnal flux of $\mathbf{F}_\gamma$ through an isopycnal surface $\gamma(S,\theta) = \text{constant}$ is given by:

$$\mathbf{F}_\gamma \cdot \frac{\nabla\gamma}{|\nabla\gamma|} = -\left[ K_H \sin^2(\nabla\gamma,\mathbf{k}) + K_V \cos^2(\nabla\gamma,\mathbf{k}) \right] |\nabla\gamma| = -\left[ (K_H - K_V)\sin^2(\nabla\gamma,\mathbf{k}) + K_V \right] |\nabla\gamma|, \tag{2}$$

where $(\nabla\gamma,\mathbf{k})$ is the angle between the local gradient of $\gamma$ and the vertical direction. This expression shows that the actual diapycnal mixing experienced by the density-like variable $\gamma(S,\theta)$ can be written as the sum $K_V + K_V^{Veronis}$, with:

$$K_V^{Veronis} = (K_H - K_V)\sin^2(\nabla\gamma,\mathbf{k}) \approx K_H \sin^2(\nabla\gamma,\mathbf{k}), \tag{3}$$

when $K_H \gg K_V$ as often assumed in ocean models. To the extent that it is legitimate to regard $K_V$ as related to measured values of diapycnal/vertical mixing, it is generally assumed that $K_V^{Veronis}$ induces spurious diapycnal mixing whenever it exceeds $K_V$, which in general occurs whenever isopycnal slopes become large enough. Rotated diffusion tensors, e.g., Redi (1982); McDougall and Church (1986), were introduced as a more natural and physical way to account for the 7 orders of magnitude difference between isopycnal and diapycnal mixing, and hence as a way to avoid the occurrence of the Veronis effect. The extent to which the reduction of spurious upwelling can truly be attributed to the introduction of rotated diffusion tensors is unclear however, as several studies suggest that this reduction should in fact be attributed to the parameterisation of meso-scale eddy induced advection, which was introduced simultaneously with parameterisation of rotated diffusion, e.g., Böning et al. (1995); Lazar et al. (1999); Huck et al. (1999).

In absence of an unambiguous definition of density for a nonlinear equation of state, rotated diffusion tensors have traditionally relied on the use of the so-called local neutral vector

$$\mathbf{N} = g\left(\alpha\nabla\theta - \beta\nabla S\right) \tag{4}$$

with $\theta$, $S$ respectively the potential temperature and the salinity and $\alpha$, $\beta$ respectively the thermal contraction and haline expansion coefficient, e.g., McDougall et al. (2014). A conceptual difficulty with neutral rotated diffusion tensors, however, is that it is not possible to construct for the ocean a mathematically well defined materially conserved variable $\gamma(S,\theta)$ allowing to write $\mathbf{N} = C_0\nabla\gamma$, with $C_0$ some integrating factor, which mathematically arises from the non-zero helicity of $\mathbf{N}$. One instructive way to show this is by assuming that such a variable $\gamma$ exists, and to show that it leads to a contradiction. To proceed, let us express in-situ density $\rho = \rho(S,\theta,p) = \hat{\rho}(\gamma,\theta,p)$ as a function of $\gamma$, $\theta$ and $p$ for instance following Tailleux (2016b). The expression for the neutral vector becomes:

$$\mathbf{N} = -\frac{g}{\rho}\left(\frac{\partial\hat{\rho}}{\partial\gamma}\nabla\gamma + \frac{\partial\hat{\rho}}{\partial\theta}\nabla\theta\right) \tag{5}$$

where:

$$\frac{\partial\hat{\rho}}{\partial\theta} = \frac{1}{J}\frac{\partial(\gamma,\rho)}{\partial(S,\theta)} \tag{6}$$





where $J = \partial(\gamma, \theta)/\partial(S, \theta) = \partial\gamma/\partial S$ is the Jacobian of the transformation going from $(S, \theta)$ to $(\gamma, S)$ space. For $\gamma$ to be exactly neutral would require $\partial\hat{\rho}/\partial\theta = 0$ everywhere, but Eq. (6) shows that this is impossible. Indeed, for $\partial(\gamma, \rho)/\partial(S, \theta)$ to be zero would require $\rho$ to be a function of $\gamma(S, \theta)$ alone, but this cannot be true, because $\rho$ also depends on pressure. This implies that the diapycnal diffusivity experienced by any mathematically well defined density variable must at least be partly controlled by

isoneutral mixing, in a way that depends on the degree of non-neutrality of the density variable considered. Mathematically, the problem arises because the local concept of neutral mixing cannot be extended globally. This idea is not entirely new, as it is closely connected to the concept of fictitious mixing discussed by McDougall and Jackett (2005) or Klocker et al. (2009) for instance. Physically, however, the concepts of effective diffusive mixing considered in the present paper and that of fictitious mixing are radically different and have different purposes and implications. Indeed, the concept of fictitious mixing

aims to quantify the extra diapycnal mixing that is potentially introduced by rotating the mixing directions along that defined by a globally defined variable $\gamma(S, \theta)$ instead of the neutral directions, without changing the isoneutral and dianeutral mixing coefficients. In contrast, the concept of effective diffusivity aims to quantify the actual — as opposed to fictitious — diapycnal mixing experienced by a given globally defined material density variable $\gamma(S, \theta)$ acted upon by neutral rotated diffusion. The concept of effective diffusivity plays a key role in the theory of water masses, as the latter is most naturally formulated in

terms of a globally defined material density variable (note, however, Iudicone et al. (2008)'s attempt to use $\gamma^n$), as well as in modern approaches to estimating spurious numerical diapycnal mixing Griffies et al. (2000); Ilıcak et al. (2012). From a mathematical viewpoint, global inversions can only give us access to the effective diffusivity associated to a given density variable $\gamma$; it is impossible to directly estimate dianeutral mixing, which must in practice be disentangled from the part of the effective diffusivity controlled by isoneutral mixing. Likewise of estimates of spurious numerical diapycnal mixing when a

realistic nonlinear equation of state is used. The idea that the effective diffusivity might be contaminated to some degree by isoneutral mixing was hypothesised by Lee et al. (2002), but they assumed the effect to be second order and made no attempt at quantifying it. Doing so is one of the main objective of this paper, which appears to be attempted here for the first time.

     For clarity, we call dianeutral and isoneutral the directions parallel and perpendicular to the local neutral tangent plane, and diapycnal and isopycnal the directions perpendicular and parallel to isopycnal surface $\gamma = constant$ defined by the particular

density variable $\gamma$ considered. As mentioned above, the idea that the mixing directions must align with the isoneutral and dia-neutral directions combined with the impossibility of constructing an exactly neutral density variable is potentially important to estimate the actual dianeutral mixing using water masses theory and to estimate spurious numerical diapycnal mixing, as is expended further below.

     Regarding the first application, it takes its root in the water mass framework originally presented by Walin (1982), whose aim

is to link surface heat fluxes to diffusion across isotherms in the interior. This work has been generalized to link the diapycnal diffusive flux to diabatic forcing of potential density at the surface by Speer and Tziperman (1992), but the theory can be easily extended to use any potential density variable. The isoneutral mixing contribution to diapycnal mixing depends on the degree of non-neutrality of the density variable $\gamma$ considered. Because exactly neutral surfaces do not exist, it is not possible to unambiguously estimate the dianeutral diffusion using a Walin-type methodology, for the result will always be biased by a

$\gamma$-dependent amount of isoneutral mixing. It is thus important to assess the degree of contamination of diapycnal mixing es-



timates by isoneutral mixing before one is able to conclude on the discrepancy between measured values of diapycnal mixing and values inferred from global budgets.

Regarding the second application, it concerns attempts at diagnosing spurious numerical mixing in numerical ocean models by means of the APE framework discussed by Winters et al. (1995) and Winters and D'Asaro (1996) (WN hereafter). Interest

in this approach is motivated by the fact that WN's APE framework has become the accepted standard as the most rigorous approach to diagnosing diapycnal mixing in the study of turbulent stratified fluids. Physically, WN's approach relies on the idea that only diapycnal mixing can cause modifications of the so-called Lorenz reference state, that is, the state of minimum potential energy obtained by means of an adiabatic re-arrangement of the fluid parcels. Such a method has been used for instance in Griffies et al. (2000) and more recently in Hill et al. (2012) and Ilıcak et al. (2012) in order to compare the

numerical diapycnal mixing associated with different numerical schemes in spin-down experiments. There is no question that monitoring the evolution of Lorenz reference state represents an exact and rigorous approach to diagnosing real or spurious diapycnal for a linear equation of state (as done in Griffies et al. (2000); Hill et al. (2012) and most of Ilıcak et al. (2012)). However, this is questionable for a binary fluid with a nonlinear equation of state such as seawater for several reasons. First, the nonlinearities of the equation of state for seawater introduce additional sinks and sources of density linked to cabelling and

thermobaricity, while also complicating the identification of the mixing directions. Second, as pointed in Tailleux (2016a), it is arguably the materially conserved property of density (the fact that it is a function of $\theta$ and $S$ alone) that is really the key feature that is used in WN's APE framework to diagnose diapycnal mixing, not its link to APE. Indeed, for a binary fluid, there is an infinite number of density variables: $\gamma(S, \theta)$, each of which can be used for diagnosing the effect of diabatic mixing processes. The density variable linked to the Lorenz reference state is a particular case of the general density variable $\gamma(\theta, S)$

and diagnosing diapycnal mixing with this Lorenz density variable in a realistic ocean with a nonlinear equation of state can only give us access to the effective diapycnal diffusivity across surfaces of constant Lorenz density. In what follows, we call this "Lorenz density variable" the reference density $\rho_{ref}(\theta, S)$, which is a function of $\theta$ and $S$ alone, the density of $(\theta, S)$ at a pressure $p_{ref} = |z_{ref}|g\rho_0$ with $g = 9.81$ m$^2$/s, $\rho_0 = 1027$ $kg/m^3$ and $z_{ref}$ the Lorentz reference detph as defined in WN for a temperature only fluid or in Saenz et al. (2015) for a binary fluid. Note that diagnosing the diapycnal diffusivity with Lorenz

reference state is by definition the same as diagnosing the total flux through a $\rho_{ref}$ surface. The departure from neutrality of $\rho_{ref}$ implies that its effective diapycnal diffusivity is partly controlled by isoneutral mixing, and hence that it would be wrong to interpret the effective diapycnal diffusivity inferred from WN's APE approach only in terms of spurious numerical mixing (without speaking of sinks and sources of density due to the non-linear equation of state) which does not appear to have been realised previously.


The main purpose of this paper is to quantify the degree of contamination of estimates of diapycnal mixing by isoneutral mixing for a number of density variable of the form $\gamma(S, \theta)$, illustrated for the following five density variables: Jackett and McDougall (1997) $\gamma^n$, three potential density variables $\sigma_0$, $\sigma_2$, $\sigma_4$ and Lorentz reference state density $\rho_{ref}$ (WN). Note that although $\omega$ surfaces Klocker et al. (2009) are more neutral than $\gamma^n$, they are likely to be less material (a material density variable

is a variable conserved whenever $\theta$ and $S$ are both conserved i.e. a function of $\theta$ and $S$ only) because neutrality is likely to





be improved at the expense of materiality. Moreover, no density variable associated with $\omega$-surfaces has been constructed yet, which makes the use of the latter impractical for the present purposes. These density variables have been chosen because they are widely used in the oceanographic community and thus deserve special attention. Section 2 presents the theoretical framework used for defining effective diffusivities for each variable. Section 3 discusses the results obtained for the above

5 mentioned 5 density variables. Finally, Section 4 summarises and discusses the results.

## 2 Method

### 2.1 Effective diffusivity

Thermodynamic properties in numerical ocean models are commonly formulated in terms of $\theta$ and $S$, whose evolution equations can in general be expressed as:

$$\frac{D_{\mathrm{res}}\theta}{Dt} = \nabla \cdot (\mathbf{K}\nabla\theta), \qquad \frac{D_{\mathrm{res}}S}{Dt} = \nabla \cdot (\mathbf{K}\nabla S), \tag{7}$$

where $\mathbf{K} = K_i(\mathbf{I} - \mathbf{dd}^T) + K_d\mathbf{dd}^T$ is the neutral rotated diffusion tensor, with $K_i$ and $K_d$ being the isoneutral and dianeutral turbulent mixing coefficients respectively, $\mathbf{d} = \mathbf{N}/|\mathbf{N}|$ the locally-defined normalised neutral vector, and $D_{\mathrm{res}}/Dt = \partial/\partial t + (\mathbf{v} + \mathbf{v}_{gm})\cdot\nabla$ the advection by the residual velocity (the sum of the resolved Eulerian velocity plus the meso-scale eddy induced velocity). As a result, the evolution equation of any material density variable $\gamma(S,\theta)$ must be given

$$\frac{D_{res}\gamma}{Dt} = \nabla \cdot (\mathbf{K}\nabla\gamma) - \underbrace{\left(\gamma_{\theta\theta}\nabla\theta^T\mathbf{K}\nabla\theta + 2\gamma_{S\theta}\nabla S^T\mathbf{K}\nabla\theta + \gamma_{SS}\nabla S^T\mathbf{K}\nabla S\right)}_{NL}. \tag{8}$$

Unless $\gamma(S,\theta)$ is a linear function of $S$ and $\theta$, its evolution equation will in general contain non vanishing nonlinear terms (denoted NL in Eq. (8)) related to cabelling and thermobaricity, e.g., McDougall (1987); Klocker and McDougall (2010); Urakawa et al. (2013). In several previous studies, it has been common to include the nonlinear terms NL as part of the definition of effective diffusivity, e.g., Lee et al. (2002). In this paper, however, we exclude the nonlinear terms from our

20 definition of effective diffusivity, and hence define the diffusive flux of $\gamma$ as:

$$F_{\mathrm{diff}}^{\gamma} = -\mathbf{K}\nabla\gamma = -\left(K_i(\nabla\gamma - (\nabla\gamma \cdot \mathbf{d})\mathbf{d}) + K_d(\nabla\gamma \cdot \mathbf{d})\mathbf{d}\right) \tag{9}$$

We define the effective diffusive flux of $\gamma$ as the integral of the diffusive flux across the isopycnal surface $\gamma(\mathbf{x},t) = \mathrm{constant}$, viz.,

$$F_{\mathrm{eff}} = -\int\limits_{\gamma=\mathrm{const}} \mathbf{K}\nabla\gamma \cdot \mathbf{n}\,\mathrm{d}S \tag{10}$$





where $\mathbf{n} = \frac{\nabla\gamma}{|\nabla\gamma|}$ is the unit local normal vector to the $\gamma$ surface. Now, it is easily established after some straightforward algebra that

$$
\begin{aligned}
K\nabla\gamma \cdot \mathbf{n} &= [K_i(\nabla\gamma - (\nabla\gamma \cdot \mathbf{d})\mathbf{d}) + K_d(\nabla\gamma \cdot \mathbf{d})\mathbf{d}] \cdot \frac{\nabla\gamma}{|\nabla\gamma|} \\
&= \left[K_i\left(|\nabla\gamma|^2 - (\nabla\gamma \cdot \mathbf{d})^2\right) + K_d(\nabla\gamma \cdot \mathbf{d})^2\right]/|\nabla\gamma| \\
&= |\nabla\gamma|\left[K_i\sin^2(\nabla\gamma,\mathbf{d}) + K_d\cos^2(\nabla\gamma,\mathbf{d})\right].
\end{aligned}
\tag{11}
$$

Eq. (11) establishes that the locally defined effective diapycnal diffusivity experienced by the density variable $\gamma$ is affected by

both isoneutral and dianeutral mixing, the contribution from isoneutral mixing being akin to a Veronis-like effect, as discussed in Tailleux (2016b). Because we are primarily interested in the latter effect, we shall discard the effect of dianeutral mixing on the effective diapycnal diffusivity of $\gamma$ and hence assume $K_d = 0$ in the rest of the paper. As a result, the expression for the effective diffusive flux of $\gamma$ becomes:

$$
F_{\mathrm{eff}} = -\int\limits_{\gamma=\mathrm{const}} |\nabla\gamma| K_i \sin^2(\nabla\gamma,\mathbf{d})\mathrm{d}S.
\tag{12}
$$

Note that the integrand of (12) is mathematically equivalent to what McDougall and Jackett (2005) refer to as "fictitious diapycnal mixing". However, here the integrand is integrated on $\gamma$ surfaces and then used to calculate an effective diffusivity coefficient which is easier to interpret than a collection of local values of the $(\nabla\gamma, \mathbf{d})$ angle.

## 2.2 Reference Profile

In order to construct an effective turbulent diffusivity $K_{\mathrm{eff}}$ associated with the effective diffusivity flux $F_{\mathrm{eff}}$, we need to define

an appropriate mean gradient for the density variable $\gamma$. This is done by constructing a reference profile for $\gamma$, as explained in the next paragraph.

Let $z_r(\gamma, t)$ be the reference profile for the particular material density $\gamma(S, \theta)$ (which can always be written as a function of space $\mathbf{x}$ and time $t$ as $\gamma^*(\mathbf{x}, t) = \gamma(S, \theta)$), constructed to be the implicit solution of the following problem:

$$
\int\limits_{V(z_r)} \mathrm{d}V = \int\limits_{V(\gamma,t)} dV = \int\limits_{z_r(\gamma,t)}^{0} A(z)dz,
\tag{13}
$$

where $A(z)$ is the depth-dependent area of the ocean at depth $z$, and $V(\gamma, t)$ the volume of water for all parcels with density $\gamma_0$ such that $\gamma_{min} \leq \gamma_0 \leq \gamma$, where $\gamma_{min}$ is the minimum value of $\gamma$ encountered in the ocean. The knowledge of the reference profile allows one to regard the volume $V(\gamma, t)$ of water masses with density lower than $\gamma$ either as a function of $z_r$ only as $V(z_r)$ so that $V(\gamma, t) = V(z_r(\gamma, t))$. Physically, Eq. (13) defines the reference depth $z_r(\gamma, t)$ so that the volume of water with density lower than $\gamma$ is equal to the volume of water comprised between the ocean surface and $z_r$; this definition is equivalent

to that used by Winters and D'Asaro (1996) or Saenz et al. (2015) to construct Lorenz reference state, but generalised here to the case of an arbitrary materially conserved density variable $\gamma(S, \theta)$. Once $z_r(\gamma, t)$ is constructed, it can be inverted to define in turn the reference profile $\gamma_r(z_r, t)$. Indeed, by definition $\gamma_r(z_r(\mathbf{x}, t), t) = \gamma^*(\mathbf{x}, t)$. As a result, we can always write a relation

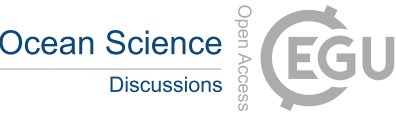

such as:

$$\nabla\gamma = \frac{\partial\gamma_r}{\partial z_r}\nabla z_r \tag{14}$$

A major difference with Winters and D'Asaro (1996) or Griffies et al. (2000) is that our definition of reference depth and density is not restricted to Lorenz reference state, for it can be applied to any arbitrary $\gamma(S,\theta)$. However, the choice of $\gamma(\theta,S)$ influences the local projection of the iso-dianeutral diffusion on the $\gamma$ gradient and thus the effective diapycnal coefficient. We now define the effective diffusivity $K_{\mathrm{eff}}$. Using (14) in (12), we get:

$$F_{\mathrm{eff}} = -\int_{\gamma=\mathrm{const}} |\nabla\gamma|K_i\sin^2(\nabla\gamma,\mathbf{d})dS = \frac{\partial\gamma_r}{\partial z_r}\int_{z_r=\mathrm{const}} |\nabla z_r|K_i\sin^2(\nabla z_r,\mathbf{d})\mathrm{d}S = A(z_r)K_{\mathrm{eff}}\frac{\partial\gamma_r}{\partial z_r}, \tag{15}$$

where we have used $|\nabla\gamma| = -\frac{\partial\gamma_r}{\partial z_r}|\nabla z_r|$ (because $\frac{\partial\gamma_r}{\partial z_r} < 0$) and where $K_{\mathrm{eff}}$ is defined by the following relation:

$$K_{\mathrm{eff}}(z_r) = \frac{\int_{z_r=\mathrm{const}} K_i|\nabla z_r|\sin^2(\nabla z_r,\mathbf{d})\mathrm{d}S}{A(z_r)}, \tag{16}$$

and is independent of the gradient of $\gamma_r$ in the reference space. $K_{\mathrm{eff}}$ is not the surface average of the local mixing coefficient across $\gamma = \mathrm{const}$. surfaces but rather the mixing coefficient linked to the time variation of $\gamma_r$ as can be seen from the following equation (a proof is shown in the appendix):

$$\frac{\partial\gamma_r}{\partial t} = \frac{1}{A(z_r)}\frac{\partial}{\partial z_r}\left(A(z_r)K_{\mathrm{eff}}(z_r)\frac{\partial\gamma_r}{\partial z_r}\right) + \mathrm{NL} + \mathrm{F} \tag{17}$$

where NL is a term due to the non linearity of $\gamma(S,\theta)$ and $F$ is a term due to the heat and haline fluxes at the ocean surface. Note that in Speer (1997) and in Lumpkin and Speer (2007), the effective diffusivity is defined as the integral of the local diapycnal flux on a $\gamma$ surface over the integral of the local gradient of $\gamma$ on the same $\gamma$ surface i.e.:

$$K_{\mathrm{eff}}^{\mathrm{speer}} = \frac{\int_{z_r=\mathrm{const}} K\nabla\gamma\cdot\mathbf{n}dS}{\int_{z_r=\mathrm{const}} \nabla\gamma\cdot\mathbf{n}dS} \tag{18}$$

is different from our formulation because of the different mean gradient formulation. The relationship between the $K_{\mathrm{eff}}$ described in this article (a generalization of Winters and D'Asaro (1996)'s formulation) and $K_{\mathrm{eff}}^{\mathrm{speer}}$ is, from formula (16) and (18):

$$K_{\mathrm{eff}} = K_{\mathrm{eff}}^{\mathrm{speer}}\left(\frac{\int_{z_r=\mathrm{const}} |\nabla z_r|dS}{A(z_r)}\right). \tag{19}$$

We have checked that for all the density variables under consideration here the quantity between brakets in (19) is smaller than 1 so that $K_{\mathrm{eff}}$ can be seen as a lower bound of $K_{\mathrm{eff}}^{\mathrm{speer}}$. In Lee et al. (2002), the effective diapycnal coefficient formulation is similar to Speer (1997)'s except that the mean gradient is approximated by an average of the vertical gradient of $\gamma$ on a $\gamma$ surface which is valid as long as the $\gamma$ slope is small.




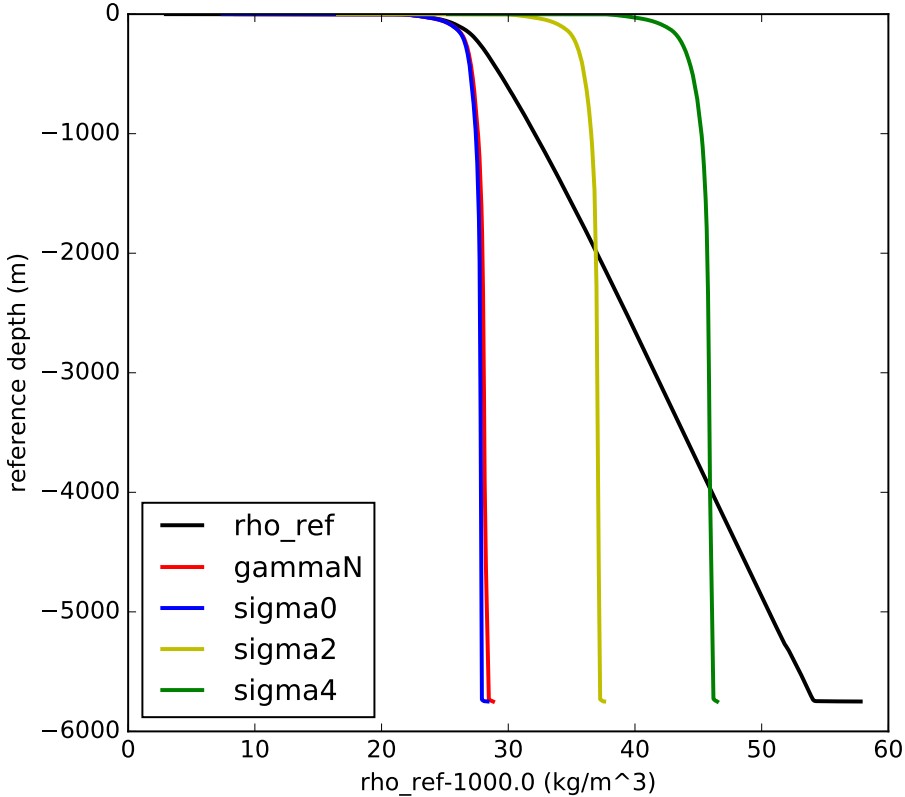

**Figure 1.** Reference density for $\rho_{ref}$ (black) $\gamma^n$ (red), $\sigma_0$ (blue), $\sigma_2$ (yellow) and $\sigma_4$ (green) as a function of the reference depth.

## 3 Isoneutrally-controlled effective diapycnal diffusivities for $\sigma_0$, $\sigma_2$, $\sigma_4$, $\gamma^n$ and $\rho_{ref}$

In this section we seek to estimate the effective diffusivity (16) derived in the previous section for five different density variables: $\sigma_0$, $\sigma_2$, $\sigma_4$, the Jackett and McDougall (1997)'s $\gamma^n$ and the Lorentz reference density $\rho_{ref}$ obtained with Saenz et al. (2015) method. All the calculation of this section are performed with annual mean potential temperature and salinity data from the World Ocean Circulation Experiment (Gouretski and Koltermann, 2004). Since $\gamma^n$ is not well defined North of 60° N, the latter region was excluded from our analysis for all five density variables. Since eddies mix the fluid horizontally in the mixed layer rather than perpendicular to the neutral vector, we also restrict our calculation to the ocean below the mixed layer. The depth of the mixed layer is given by the de Boyer Montégut database (de Boyer Montégut et al., 2004). The reference density for each of the five variables is shown on figure 1. As expected, the range of values taken by the reference density of the three potential density variables increases with the reference pressure. $\gamma^n$ has a reference density similar to that of $\sigma_0$ with a slightly smaller gradient in the reference space. $\rho_{ref}$ has a gradient much smaller than all other density variables. It crosses $\sigma_0$ at the surface, $\sigma_2$ around $-2000$ meters and $\sigma_4$ around $-4000$ meters. This is due to the fact that the volume above the sur-




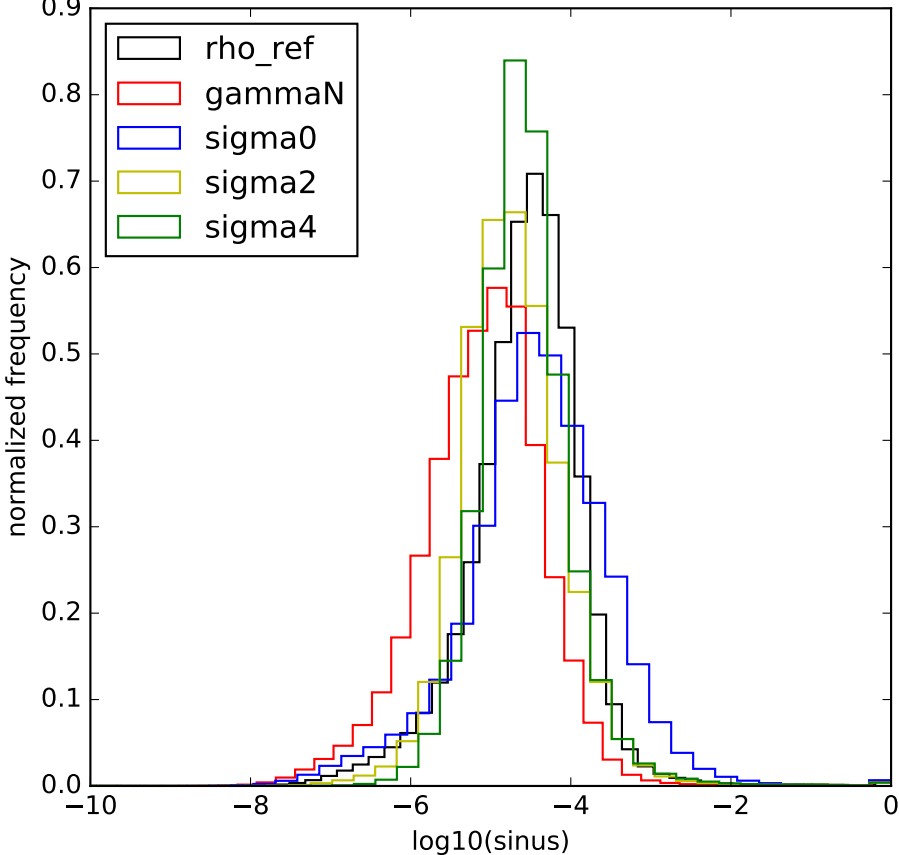

**Figure 2.** histogram of the decimal logarithm of the squared sinus between the gradient of $\gamma$ and the neutral vector $\mathbf{d}$ weighted by the volume of each point. $\log_{10}\left(\sin^2\left(\nabla\gamma, \mathbf{d}\right)\right)$ for $\rho_{ref}$ (black), $\gamma^n$ (red), $\sigma_0$ (blue), $\sigma_2$ (yellow) and $\sigma_4$ (green)

face $\sigma_p(\theta, S) = \sigma_p^r(Z)$ is by definition the same as the volume above $\rho(\theta, S, p) = \rho_{ref}(Z)$ where $p = -Z\rho_0 g$ is the reference pressure linked to the reference depth $Z$, $\sigma_p^r$ is the reference density linked to $\sigma_p$.

Figure 2 shows the histogram of the decimal logarithm of the squared sinus of the angle between $\nabla\gamma$ and $\mathbf{d}$ ( calculated using formula A1) shown in appendix A): $\log_{10}[\sin^2(\nabla\gamma, \mathbf{d})]$ and weighted by the volume associated with each point. This plot is

5    similar to that discussed by McDougall and Jackett (2005) in their discussion of fictitious diapycnal mixing.

$\rho_{ref}$, $\sigma_2$ and $\sigma_4$ give similar angles with most of their values slightly larger than $10^{-5}$. $\gamma^n$ gives the smallest angles among the variables under consideration here with most of its values smaller than $10^{-5}$ while $\sigma_0$ gives the largest with a large number of points with values larger than $10^{-4}$. All together, these observations could suggest that the effective diffusivity of $\gamma^n$ should be the smallest overall, that the effective diffusivity of $\rho_{ref}$ should be of the same order as that for $\sigma_2$ and $\sigma_4$, and that the effec-

10    tive diffusivity for $\sigma_0$ should be the largest of all. It is however hard to predict the values of the effective diffusivity coefficient



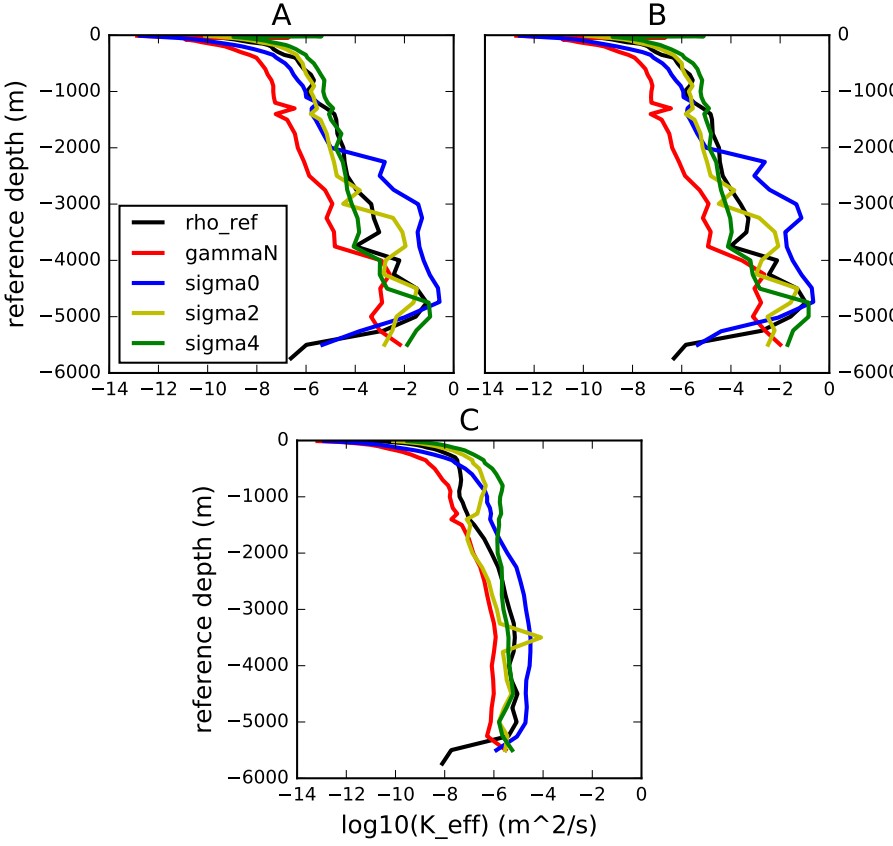

**Figure 3.** $\log_{10}$ of the effective diapycnal diffusivity coefficient $K_{\text{eff}}$ as a function of the reference depth (meters) (and as defined by equation (16)) for $\rho_{ref}$ (black), $\gamma^n$ (red), $\sigma_0$ (blue), $\sigma_2$ (yellow) and $\sigma_4$ (green). Each panel correponds to a $K_{\text{eff}}$ calculated with different isoneutral diffusivity coefficient. A: $K_{iso} = 1000 \text{ m}^2/\text{s}$, B: variable isoneutral diffusivity coefficient given by Forget et al. (2015). Bottom same as B but without 5% of the largest angles (C)

for each density variable from figure 2 only since the small amount of point with very large angle values (hardly visible on figure 2) could overcome the large amount of points with small angles and since the spatial variability of the isoneutral mixing coefficient could correlates with the spatial variability of the angle. We thus calculate the effective diffusivity coefficient using these angles values for each density variable.

Figure 3 shows the decimal logarithm of the effective diffusivity $K_{\text{eff}}$ for the five variables as a function of the reference depth under two possible choices of $K_i$:



The first case (A, figure 3) assumes a constant isoneutral coefficient: $K_i = 1000$ m$^2$/s. Under this assumption, $K_{\text{eff}}$ for every density variables increases on average with the reference depth from values between $10^{-12}$ and $10^{-8}$ m$^2$/s close to surface reference depth to values between $10^{-6}$ and $0$ m$^2$/s at the deepest reference depths. This increase can be attributed to the fact that the largest discrepancy between the neutral vector and the gradients of the 5 density variables is generally located in the

ACC (Antarctic Circumpolar Current) (as will be shown later) where the highest densities, and thus deepest reference depths, outcrop.

$K_{\text{eff}}$ for $\gamma^n$ and $\sigma_0$ are similar between 0 and 800 m depth with values ranging from $10^{-8}$ m$^2$/s at the surface to $10^{-6}$ m$^2$/s at -800 meters. $\sigma_2$, $\sigma_4$ and $\rho_{ref}$ give values up to 100 larger on the same depth range. Between 800 and 4000 m depth, $\gamma^n$ gives the smallest $K_{\text{eff}}$ which is slowly increasing from $10^{-6}$ to $10^{-5}$ m$^2$/s as the depth decreases. On the same depths, $\rho_{ref}$, $\sigma_0$, $\sigma_2$

and $\sigma_4$ gives values at least 10 times larger (up to 1000 times larger for $\sigma_0$ below -2000 m). Below 4000 m depth, all density variables gives $K_{\text{eff}}$ larger than $10^{-4}$ m$^2$/s. At the deepest levels, under -5000 meters, $\sigma_0$ and $\rho_{ref}$ give smaller $K_{\text{eff}}$ than $\gamma^n$.

The second case (B, figure 3) assumes a spatially variable isoneutral coefficient given by the inverse calculation of Forget et al. (2015), which gives a three dimensional distribution of $K_i$ at about $1°$ resolution for the global ocean. This database contains values ranging from 9000 m$^2$/s (in the Atlantic deep water formation zone at the surface, in western boundary currents and

ACC) to values close to 0 (in the deep pelagic ocean). The estimated $K_{\text{eff}}$ for this choice are very close to those obtained under the previous assumption of constant diffusivity for all variables, showing the small sensitivity of our results to spatial variations of isoneutral diffusion. The following calculations are based on the use of a spatially varying $K_i$.

To investigate the importance of the localised large departure from neutrality in the construction of $K_{\text{eff}}$, we removed 5% of the largest non-neutral values of the angle for each reference surface (figure 3, case C). Without 5% of the largest values, $K_{\text{eff}}$

is much smaller than the previous one for every density variables with values everywhere smaller than $10^{-4}$ m$^2$/s. As before, the effective diffusivity increases rapidly close to the surface and then more slowly below -1000 meters (except at a few depth for $\sigma_2$, $\sigma_4$ and at deep reference depth for $\rho_{ref}$ and $\sigma_0$) with the reference depth for all density variables. $\gamma^n$ gives the smallest values for almost all reference depths, with values from $10^{-10}$ m$^2$/s close to the surface of the reference space to $10^{-6}$ m$^2$/s at the deepest levels. $\sigma_2$ gives the second smallest values for reference depths smaller than -1500 meters but is overtaken by

$\sigma_0$ and $\rho_{ref}$ at larger depths. $\rho_{ref}$, $\sigma_0$, $\sigma_2$ and $\sigma_4$ all give effective diffusivities of the order or larger than $10^{-5}$ m$^2$/s at some depth below -2000 meters.

This calculation shows that the isoneutral contribution to effective diapycnal mixing is very localised spatially with 5% of each surface accounting for most of the effective diffusivity for all the density variables under consideration here. However, even without this top 5%, $K_{\text{eff}}$ remains close or above $10^{-5}$ m$^2$/s for all variables except $\gamma^n$.


Figure 4 shows a meridional section of the decimal logarithm of the sinus in the Atlantic for $\rho_{ref}$, $\gamma^n$ and $\sigma_0$. The regions where the angle between the neutral vector and the gradient of the density variable is large are found mostly in the ACC at all depth for $\rho_{ref}$ and $\gamma^n$ and everywhere at depth for $\sigma_0$, suggesting that, in this region, all the density variables studied above introduce significant biases in the estimation of diapycnal mixing.





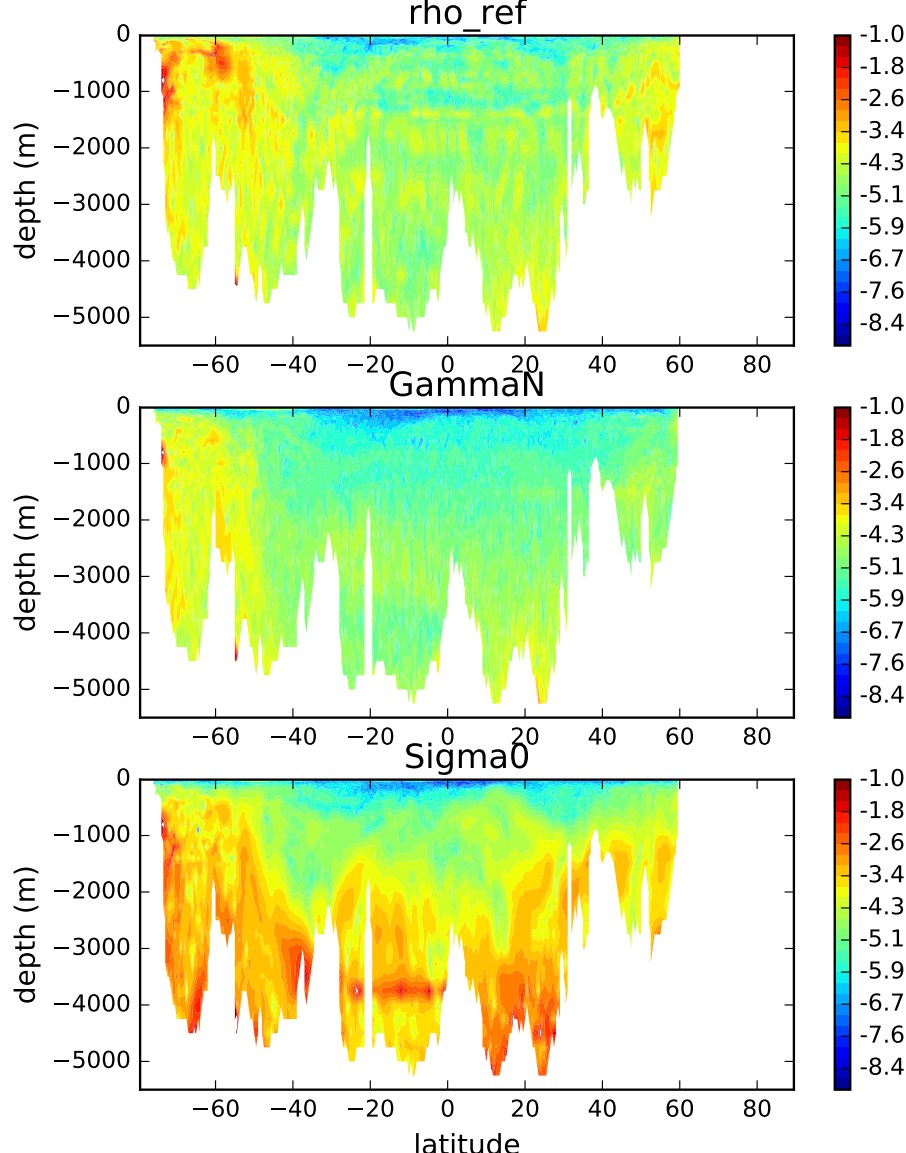

**Figure 4.** Decimal logarithm of the sinus between the neutral vector and the gradient of $\rho_{ref}$ (top), $\gamma^n$ (middle) and $\sigma_0$ (bottom) as a function of latitude and depth at 330 of longitude (in the Atlantic).

## 4   Conclusions

In this paper, we have presented a new framework for assessing the contribution of isoneutral diffusion to the effective diapycnal mixing coefficient $K_{\mathrm{eff}}$ for five different density variables, chosen for their widespread use in the oceanographic community,



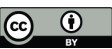

namely $\gamma^n$, $\rho_{ref}$, $\sigma_0$, $\sigma_2$, $\sigma_4$. Our results reveal that, due to the projection of the isoneutral mixing on the diapycnal direction, the actual diapycnal mixing experienced by each density variable can reach values as high as $10^{-4}$ m$^2$/s and up to 1 m$^2$/s for reference depths deeper than -2000 meters. As expected, $\gamma^n$, constructed to be as neutral as practically feasible, is the least affected by isoneutral diffusion among all density variables considered. Nevertheless, it still appears to experience values larger

than $10^{-4}$ m$^2$/s for reference depths below -4000 meters. An added difficulty pertaining to the use of $\gamma^n$, not discussed in this paper, stems from its non-material character. As a result, the validity of defining an effective diapycnal diffusivity for $\gamma^n$ using the present approach depends on such non-material effects to be small, or at least much smaller than the contribution from isopycnal diffusion discussed here, which is difficult to evaluate.

Our results thus suggest that the potential contamination due to isoneutral mixing should always be assessed for any inference of

diapycnal mixing based on the use of any density variable $\gamma(S, \theta)$ in Walin-like water mass analysis for instance. In agreement with previous studies such as McDougall and Jackett (2005), the regions of large discrepancy between the neutral vector and the gradient of each surface are very localised in space. However, while representing a very small amount of volume of the ocean, these discrepancies are important in setting the effective diffusivity values. Indeed, only 5% of the largest values on each reference surface explain of the estimated effective diffusivity coefficients. Without these 5%, none of the variables gives

a coefficient larger than $10^{-4}$ m$^2$/s. The concentration of discrepancies is even stronger for $\gamma^n$ since the effective diffusivity coefficient after the removal of the 5% of the largest values decreases below $10^{-6}$ m$^2$/s.

Similarly, our results show that the evaluation of effective diapycnal mixing using a sorting algorithm of density (e.g. Griffies et al. (2000); Ilıcak et al. (2012)), which amounts to diagnosing the diapycnal flux through $\rho_{ref}$, is likely to be significantly contaminated by isoneutral diffusion owing to the large departures from neutrality of $\rho_{ref}$ in the polar regions if a nonlinear

equation of state is used. Note that this is a distinct effect from the density sinks and sources due to the non-linear equation of state influencing the time variation of the reference density (see equation (17)) which are also a source of contamination of the diapycnal flux from the isoneutral diffusion when using sorting algorithm. It follows that diagnosing the spurious diapycnal mixing resulting from numerical advection schemes for a nonlinear equation of state remains an outstanding challenge, and that progress on this topic must take into account the theoretical considerations developed here.

This work advocates for the construction of a density function $\gamma(\theta, S)$ that would minimizes the isoneutral influence on the effective diapycnal diffusivity coefficient. So far, the best material density variable is a function of Lorenz reference density, as showed by Tailleux (2016a), but as discussed by Tailleux (2016b), it appears theoretically possible to construct an even more neutral one. Whether Klocker et al. (2009) can be used for global inversions is unclear, because its improved neutrality might be achieved at the expenses of materiality, which remains to be quantified.

In theories of the Atlantic Meridional Overturning Circulation (AMOC) (e.g. Vallis (2000); Wolfe and Cessi (2010); Nikurashin and Vallis (2011, 2012)) the diapycnal diffusion coefficient is generally assumed to be given by the dianeutral coefficient and to be of the order of $10^{-5}$ m$^2$/s. However, our results suggest that even when isopycnals are given by a density variable close to the neutral vector (e.g. with $\gamma^n$), the effective diapycnal coefficient can be much larger than the dianeutral coefficient because of the isoneutral diffusion. The issue of the amount of diapycnal mixing is an important one, as illustrated for instance by

Nikurashin and Vallis (2012) who showed that low and large diapycnal coefficient give two different regimes of the AMOC





and thus possibly two different evolution under climate change. Obviously this effect appears only when the equation of state for density is a non-linear function of both temperature and salinity we thus argue that future work should consider such non-linear equation of state for density.

**Appendix A: Numerical calculation of $\sin(\nabla\gamma, \mathbf{d})$**

To calculate the numerical value of $\sin(\nabla\gamma, \mathbf{d})$ we use the cross product between $\nabla\gamma$ and $\mathbf{d}$:

$$\sin(\nabla\gamma, \mathbf{d}) = \frac{|\nabla\gamma \times \mathbf{d}|}{|\nabla\gamma|} \tag{A1}$$

where $\times$ is the cross product. This method can be used with all the variables studied here since it only requires the knowledge of $\gamma(S, \theta)$.

**Appendix B: equation (17)**

The evolution equation for $\gamma$ is:

$$\frac{d\gamma}{dt} = \frac{\partial\gamma}{\partial\theta}\frac{d\theta}{dt} + \frac{\partial\gamma}{\partial S}\frac{dS}{dt} = \frac{\partial\gamma}{\partial\theta}\nabla(K\nabla\theta) + \frac{\partial\gamma}{\partial S}\nabla(K\nabla S) + \frac{\partial\gamma}{\partial\theta}f_\theta + \frac{\partial\gamma}{\partial S}f_S \tag{B1}$$

$$= \nabla(K\nabla\gamma) - K\nabla\theta\cdot\nabla\left(\frac{\partial\gamma}{\partial\theta}\right) - K\nabla S\cdot\nabla\left(\frac{\partial\gamma}{\partial S}\right) + f_\gamma \tag{B2}$$

where $f_\theta$, $f_S$ are the surface heat and haline fluxes and where $f_\gamma = \frac{\partial\gamma}{\partial\theta}f_\theta + \frac{\partial\gamma}{\partial S}f_S$. Then let $z_r(X, t)$ be the reference level of $\gamma$ defined by equation (13) so that $\gamma$ can now be written: $\gamma(S, \theta) = \gamma_r(z_r, t)$. Then integrating (B2) on a volume $V(z_r)$ defined

by water parcels of reference level larger than or equal to $z_r$ gives:

$$\int_{V(z_r)} \frac{\partial\gamma}{\partial t}dV + \gamma_r(z_r, t)\int_{z_r=\text{const}} \mathbf{u}\cdot\mathbf{n}dS = \int_{z_r=\text{const}} K\nabla\gamma\cdot\mathbf{n}dS - \int_{V(z_r)} K\nabla\theta\cdot\nabla(\frac{\partial\gamma}{\partial\theta}) + K\nabla S\cdot\nabla(\frac{\partial\gamma}{\partial S})dV + \int_{V(z_r)} f_\gamma dV \tag{B3}$$

where $z_r = \text{const}$ refers to the constant $z_r$ surface. $\mathbf{n} = \frac{\nabla\gamma}{|\nabla\gamma|} = -\frac{\nabla z_r}{|\nabla z_r|}$ is the local normal to the surface $\gamma = \text{const}$, the minus sign arises because the integration is done toward deeper values of $z_r$. The second term on the left hand side is zero because of the non-divergence of the velocity and the first term can be written as:

$$\int_{V(z_r)} \frac{\partial\gamma}{\partial t}dV = \frac{\partial}{\partial t}\int_{V(z_r)} \gamma_r dV' - \gamma_r \underbrace{\frac{\partial V(z_r)}{\partial t}}_{=0} \tag{B4}$$

The second term on the right hand side is zero because the total volume at constant $z_r$ is independent of time (see formula (13)). Using (B4) and the $z_r$ derivative of (B3) we get:

$$\frac{\partial\gamma_r}{\partial t} = \frac{1}{A(z_r)}\frac{\partial}{\partial z_r}\left(A(z_r)K_{\text{eff}}(z_r)\frac{\partial\gamma_r}{\partial z_r}\right) + \text{NL} + \text{forcing} \tag{B5}$$





where we have used formula (13) and the fact that the volume integral of a $z_r$ only function can be expressed as an integral over the reference depth:

$$\frac{\partial}{\partial z_r}\left(\frac{\partial}{\partial t}\int\limits_{V(z_r)}\gamma_r dV'\right) = \frac{\partial}{\partial t}\left(\frac{\partial}{\partial z_r}\int\limits_{z_r}^{0}A(z_r')\gamma_r(z_r',t)dz_r'\right) = -A(z_r)\frac{\partial\gamma_r}{\partial t} \tag{B6}$$

and with:

$$5 \quad \text{NL} = \frac{1}{A(z_r)}\frac{\partial}{\partial z_r}\left(\int\limits_{V(z_r)}\left(K\nabla\theta\cdot\nabla(\frac{\partial\gamma}{\partial\theta}) + K\nabla S\cdot\nabla(\frac{\partial\gamma}{\partial S})\right)dV\right) \tag{B7}$$

and

$$\text{forcing} = -\frac{1}{A(z_r)}\frac{\partial}{\partial z_r}\left(\int\limits_{V(z_r)}f_\gamma dV\right) \tag{B8}$$

and finally $K_{\text{eff}}$ given by formula (16).

*Acknowledgements.* This work was supported by the grant NE/K016083/1 "Improving simple climate models through a traceable and
10   process-based analysis of ocean heat uptake (INSPECT)" of the UK Natural Environment Research Council (NERC). Modeling results
presented in this study are available upon request to the corresponding author.



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
