# Peer review of "Isoneutral control of effective diapycnal mixing in numerical ocean models with neutral rotated diffusion tensors"

_Ocean Science, 2017_

## Short Comment (SC1) · 23 Sep 2017

Review of the Ocean Science submission "Isoneutral control of effective diapycnal mixing in numerical ocean models with neutral rotated diffusion tensors" by Antoine Hochet, Remi Tailleux, David Ferreira and Till Kuhlbrodt.

This paper quantifies the extent of non-neutrality for a few candidate density variables. This paper has some interesting things to say and it should be published in Ocean Science.

Throughout the paper it would be more convincing if Conservative Temperature, $\Theta$, and Absolute Salinity, $S_A$, were used in place of potential temperature and Practical Salinity, since these are the variables that have been adopted by IOC under TEOS-10 and because (1) Conservative Temperature $\Theta$ is many times more conservative than is potential temperature, and (2) Absolute Salinity $S_A$ takes into account the varying seawater composition, and in particular, the effect of this varying composition on the specific volume of seawater.

Just below Equation (7), and throughout the following equations (particularly Equation (11)), the small-scale turbulent mixing coefficient is not represented correctly. Small-scale mixing is isotropic. It does not diffuse stuff only in the diapycnal direction, but rather it diffuses stuff isotropically. McDougall et al (2014) (JPO) discuss this. The equations in this paper should be changed accordingly, in keeping with how mixing actually works in the real ocean.

Figure 2. I think that the x-axis of this figure is correctly labelled, but the caption and its description in the text (line 4 of page 9) is not correct. That is, is this axis the log of the square of the sine, or is it half this? Also, please check this issue for what is plotted in Figure 4; is it the square or not.

Page 8, Line 11. Here it says "has a gradient much smaller than all …". I think that this should read "has a gradient much larger than all …".

Page 13, line 3. "depths deeper than -2000 meters." This negative depth would be 2000 m above the sea surface, in the atmosphere. So I think what is meant is "depths deeper than 2000 meters."

---

## Referee Comment (RC1) · S. Groeskamp (Referee) · 3 Nov 2017

**Review of "Isoneutral control of effective diapycnal mixing in numerical ocean models with rotated diffusion tensors"**

Hochet et al., Ocean Science.
By Sjoerd Groeskamp, 03-11-2017.

Analysing the neutrality of density variables is an important check to understand which variables can be used in which situations. This paper expresses the non-neutrality of a given density variable as an effective mixing, meaning as a diapycnal mixing that is a result of the misalignment between the given density surface and the actual neutral direction. This paper needs to address some important issues. However, after some work, this is publishable. Below are some general and some detailed comments that will need to be addressed to make the paper better and suitable for publications.

**General Comments**

It is good to search for a materially conserved variable that is as neutral as possible, but I feel this paper oversells the need for such a variable (for example P13-L5, but also other places). Density variables are either a global surface, or they are neutral. They can't be both. For some studies, it is better to have a variable that is perfectly or near neutral, but not materially conserved. Other studies may be better off with a variable that is materially conserved, but not so neutral. This depends on the research question. So, when advocating for a conserved surface, be clear it must give up neutrality and that both kinds have a role to play in the oceanographic community and that this y(S,T) is not the holy grail.

Please specify the salinity variable. There are many different types of salinity variables and they all have a different interpretation and meaning to different communities. If it is practical Salinity, then use Sp. However, when using WOCE data, one should provide the data in Conservative Temperature and Absolute Salinity variables (that should be an editorial decision from a couple of years ago). Please use correct symbols along with it. If it is "model salinity", please be specific about that.

Please provide a simple graph that illustrates the angles and vectors discussed in the introduction, specifically regarding equation (1)-(3), 11 and 15.

I'm a bit confused about the use of a horizontal and vertical diffusion tensor (Eq. 1). Common practice is to define a isoneutral (mesoscale) and isotropic (small-scale) mixing tensor, not a horizontal and vertical one. Not for convenience, but because of the different physical processes that are related to it. This part should be rewritten to include an isotropic and isoneutral mixing tensor and relate the related physical processes. If not, it should be clearly explained here how the horizontal and vertical mixing coefficient are justified, related to the physical processes that allow for the definition of vertical and horizontal mixing instead of isoneutral and isotropic mixing coefficients.

In addition to the above, Eq (9) diffuses along a density surface. Hence I disagree with the use of Kd in this equation. This is essential small-scale mixing, which is isentropic. This means that it does not require a rotation into the diapycnal direction and should not be in

this equation. I refer to section 4 of McDougall et al 2014 for further reading. Please adapt this in Eq. 9 up until the discussion of this term below eq. 11. If this term is removed from Eq. (9), then Eq. (9) represent the gradient of Y(T,S) along the actual isoneutral direction. Of this, the component across its own surface is taken, which then provide the diapycnal transport, due to dianeutral diffusion of itself.

P3-L23-28: Defines isoneutral, isopycnal, dianuetral and diapycnal. I like that. However, it would be worth moving this to an earlier stage in the manuscript. On top of that, throughout the manuscript there are places where isoneutral, isopycnal, dianuetral and diapycnal are confused. For example,

- P3-L32: Second sentence: Isoneutral contribution to diapycnal? Should this not be isopycnal to dianeutral?
- P4-L26, Isoneutral or isopycnal?
- P4-L31, Again, isopycnal not isoneutral. According to your definition, isoneutral mixing is by definition along neutral surfaces and has no diapycnal component.

Please read carefully through this manuscript and make sure this is corrected at every point.

How exactly is the neutral direction calculated? Calculating the exact neutral direction (**d**) can be tedious. Details why so, can be found in Appendix B of Groeskamp et al (2016) and probably also in Griffies et al (1998), although I'm not 100% sure of that. This has not been clarified in this paper and I'm wondering how this is done, as this is a major part of the analyses.

**Specific Comments**

- P2-L24, reference McDougall for Helicity.
- P3-L34: I'm not sure if this statement is correct. If you choose to calculate WMT due to sigma2, but you know the neutral direction everywhere then I think you may be able to calculate the WMT due to non-neutrality by exploring the sigma2 gradient in the neutral direction. For interpretation of the WMT it is still better to have an "as neutral variable as possible", but I think the statement here is too strong.
- P4-L13: Please explain "Binary Fluid".
- P4-L16: What density are you talking about here? Density depends on pressure to. Are you here talking about y(S,T)?
- P4-L33: Neutral density is categorized as a surface of y(S,T), but it also depends on P, x and y.
- P5-L10: This Tensor is defined in Redi(1982), but was not correct for the small slope approximation (GM90). This is explained in McDougall et al (2014). Another version of this tensor is given by Griffies 1998. Few references are at place here.
- P5-L20 I'm perfectly happy if you do not include the nonlinear terms, specially because they are their own form of flux. But a little sentence arguing why these are excluded would be nice.
- P6-L20 I think it should be V(y0,t).
- P6-L20-25, this was a bit hard to understand. A simple schematic would help.
- In Fig 3., why is the black line smaller than all the others, at great depths?
- P11-L35 - Maybe not Neutral density, if you remove the 5%.

- In section 3, 10^-4 is chosen as a sort of a reference diffusivity. This seems a bit random. First of all, 10^-4, is actually very diffusive for an ocean, as in most places it is way lower. Recent inverse and observationally based methods have shown this is probably much smaller, for global values (Zika et al 2010, Groeskamp et al 2017, Waterhouse et al 2014, Lumpkin and Speer 2007, etc). Also, I'm wondering, does the error not depend on what the local rate of dissipation is? If the local rate of dissipation is 10^-3 (near topography) and the error is 10^-4 from this method, does that not mean the error is still relatively small? I suggest a deeper discussion of the use of 10^-4 and the relation to local diapycnal mixing.
- P13-L5: This discussion should include a reference to the attempt of McDougall and Jacket 2005 to quantify the terms in the material derivative of Neutral Density, as some of the terms are small. What does that mean for Neutrality?
- P13-L24: Which exact developments of this paper need to be taken into account and for what purposes exactly? This paper provides no information on how to reduce mixing by changing advection schemes. Please specify.

**Figures**

For all the figures, improved titles and perhaps some annotation, could help make them more "stand alone". This would improve your paper. A few examples:
- Fig. 1,2 and 3: add grid lines to improve readability of results.
- Fig. 3: Put "A, B, and C" in the figure itself and give clear titles.

**Citations**

- McDougall, T. J., S. Groeskamp, and S. M. Griffies (2014), On geometrical aspects of interior ocean mixing, J. Phys. Ocean., 44(8), 2164–2175, doi:10.1175/JPO-D-13-0270.1.
- Waterhouse, A. F., et al. (2014), Global patterns of diapycnal mixing from measurements of the turbulent dissipation rate, J. Phys. Ocean., 44(7), 1854–1872, doi:10.1175/JPO-D-13-0104.1.
- Zika, J. D., T. J. McDougall, and B. M. Sloyan, 2010: A tracer contour inverse method for estimating ocean circulation and mixing. J. Phys. Oceanogr., 40, 26–47
- Groeskamp, Sloyan, Zika, McDougall, (2017), Mixing Inferred from an Ocean Climatology and Surface Fluxes, JPO.
- Groeskamp, R. P. Abernathey, and A. Klocker, 2016: Water mass transformation by cabbeling and thermobaricity. Geophys. Res. Lett., 43, 10 835–10 845.
- Griffies, S. M. (1998), The Gent–Mcwilliams skew flux, J. Phys. Ocean., 28(5), 831–841

---

## Short Comment (SC2) · 16 Dec 2017

**Response to a short comment by Prof. McDougall**

We thank Prof. McDougall for his comments and generally supporting comments. A detailed response can be found below.

- **This paper quantifies the extent of non-neutrality for a few candidate density variables. This paper has some interesting things to sayand it should be published in Ocean Science.**

[Figure]

We thank the referee for his support. Note though that the way we see it, the main aim of the paper is to quantify the effective diapycnal mixing experienced by a few candidate density variables acted upon by neutral rotated diffusion, rather than quantifying the extent of their non-neutrality. Although the two issues are strongly linked, they are nevertheless quite different conceptually.

- **Throughout the paper, it would be more convincing if Conservative Temperature $\Theta$ and Absolute Salinity $S_A$ were used in place of potential temperature and practical salinity, since these are the variables that have been adopted by IOC under TEOS-10 and because (1) Conservative Temperature $\Theta$ is many times more conservative than is potential temperature, and (2) Absolute Salinity $S_A$ takes into account the varying seawater composition, and in particular, the effect of this varying composition on the specific volume of seawater**

The concept of effective diapycnal diffusivity that our paper focuses on is a priori independent of the particular thermodynamic standard chosen, so that it is unclear to us what aspect of our study could be made more convincing by a different choice of thermodynamic variables. Moreover, it seems important to stress out that our study, like the overwhelming majority of oceanographic studies, relies on the use of reference composition salinity $S_R$, which is one of the particular form of Absolute Salinity discussed in TEOS-10. The fact that the reference composition salinity $S_R$ that underlies the climatological date and software used for our computations is expressed in practical salinity units does not mean that we are not using Absolute Salinity. Indeed, the fact that $S_p$ and $S_R$ are linearly related to each other means that $S_p$ and $S_R$ are interchangeable in practice, in the same way that the use of degrees Kelvin, Celsius or Fahrenheit represent equivalent ways to express in-situ temperature. The referee's suggestion to use Density Salinity in order to account for spatial variations in composition is interesting, but arguably not feasible in ocean climate studies until numerical ocean models in-

clude prognostic equations for the other various constituents modifying seawater density at second order. From a mathematical viewpoint, a single salinity variable is not sufficient to describe seawater with varying composition, so that the equations used in numerical ocean models only make sense for reference composition salinity. Since both the referee and TEOS-10 use the term 'Absolute Salinity' in place of Density Salinity, it would be misleading for our study to use the term since it does not account for spatially varying composition. As to the recommendation of using Conservative Temperature in place of potential temperature, our understanding is that it is only a personal recommendation of Prof. McDougall that is not part of the new standard for seawater endorsed by UNESCO. Indeed, our understanding is that what UNESCO endorsed is expressing the thermodynamic properties of seawater in terms of a Gibbs function, whose natural variables are Absolute Salinity, ITS-90 absolute temperature, and pressure. As discussed in Tailleux (2015), it would be very easy to retain the non-conservation of potential temperature in the evolution equation for heat carried by numerical ocean models, thus making it as accurate as that for Conservative Temperature. Using one or the other is therefore purely a matter of personal preference, and should be determined by considerations of the added computational cost of diagnosing the non-conservation of potential temperature on the one hand, versus that of the back-and-forth conversions between Conservative Temperature and in-situ temperature and its concomitant loss of significant digits.

- **Just below Equation (7), and throughout the following equations (particularly Equation (11)), the small-scale turbulent mixing coefficient is not represented correctly. Small-scale mixing is isotropic. It does not diffusive stuff only in the diapycnal direction, but rather it diffuses stuff isotropically. McDougall et al. (2014) discuss this. The equations in this paper should be changed accordingly, in keeping with how mixing actually work in the ocean**

We do not understand this comment. Indeed, as far as we understand it, the standard rotated diffusion tensor $K = K_i(I - dd^T) + K_d dd^T$ describes isotropic mixing as it is, at least as far as the locally-reference potential density $\rho_{lr}$ is concerned, since the form of $K$ implies for the latter:

$$\frac{D\rho_{lr}}{Dt} = \nabla \cdot (K_d \nabla \rho_{lr}) + N.L., \tag{1}$$

where N.L. refers to the terms arising from cabelling and thermobaricity. As far as we are aware, small-scale mixing of potential temperature (or Conservative Temperature) and salinity ($S_R$ or $S_P$) is not isotropic, based on the study by Smith and Ferrari (2009). However, even if one were to accept the idea of isotropic mixing for $\theta$ and $S$, it would only mathematically amount to modify the above tensor as follows: $K^* = (K_i + K_d)(I - dd^T) + K_d dd^T = K_i^*(I - dd^T) + K_d dd^T$, where $K_i^* = K_i + K_d$ is a modified isoneutral turbulent mixing coefficient. Clearly, the modified diffusion tensor has exactly the same structure as the non-modified one; moreover, since $K_i$ is about 7 orders of magnitude larger than $K_d$, with both coefficients having large uncertainties, $K_i^*$ and $K_i$ are clearly indistinguishable from each other.

- **Figure 2. I think that the $x$-axis of this figure is correctly labeled, but the caption and its description in the text (line 4 of page 9) is not correct. That is, is this axis the log of the square of the sine, or is it half this? Also, please check this issue for what is plotted in Figure 4: is it the square or not.** The referee is right, it is the decimal logarithm of the sinus for figure 2 and 4 (now 3 and 5). We corrected figure 2's caption accordingly.

- **Page 8, Line 11. here it says "has a gradient much smaller than all $\cdots$" I think that this should read "has a gradient much larger than all..."** The referee is right, we have corrected this sentence.

- **Page 13, line 3. "depths deeper than -2000 meters". This negative depth would be 2000 m above the sea surface, in the atmopshere. So I think what is meant is "depths deeper than 2000 meters".**

Agreed, thanks for pointing this out.

**Rémi Tailleux and Antoine Hochet**

**References**

- Smith, K. S and R. Ferrari, 2009: The production and dissipation of compensated thermohaline variance by meso-scale stirring. J. Phys. Oceanogr., 39, 2477-2501.

- Tailleux, R., 2015: Observational and energetics constraints on the non-conservation of potential/Conservative Temperature and implications for ocean modelling. Ocean Modelling, 88, 26-37.

---

## Author Comment (AC1) · 17 Dec 2017

article [utf8]inputenc fourier gensymb [english]babel color ulem soul [protrusion=true,expansion=true]microtype amsmath,amsfonts,amsthm [pdftex]graphicx lscape url

[Figure]

**Response to Sjoerd Groeskamp's review**

Antoine Hochet and Rémi Tailleux

December 17, 2017

We thank the referee for his comments and for being generally supportive of our paper. Our response to his comments are provided below.

**General Comments**

**It is good to search for a materially conserved variable that is as neutral as possible, but I feel this paper oversells the need for such a variable (for example P13-L5, but also other places). Density variables are either a global surface, or they are neutral. They can't be both. For some studies, it is better to have a variable that is perfectly or near neutral, but not materially conserved. Other studies may be better off with a variable that is materially conserved, but not so neutral. This depends on the research question. So, when advocating for a conserved surface, be clear it must give up neutrality and that both kinds have a role to play in the oceanographic community and that this $\gamma(S,T)$ is not the holy grail.**
As the referee says, the particular type of density variable of interest depends on the research question. In the present paper, the research question is the determination of

the effective diapycnal diffusivity for density variables acted upon by a neutral rotated diffusion tensor. Since the concept of effective diffusivity is physically well defined only for a density variable that is both globally defined and exactly material, it seems only normal that we should emphasize the need for a materially conserved variable as neutral as feasible, since this is the only way to make the effective diffusivity as close as possible to the specified diapycnal diffusivity. We don't understand how our paper can oversell a property that is essential for the present purposes. If the referee thinks that this is the case, he should take advantage of the public character of the discussion to explain to the reader the benefits of density variables emphasizing neutrality over materiality, as we do not think that we are sufficiently qualified to make such a case. In any case, the referee seems to make a very subjective judgment here, as nowhere in our paper do we make the claim that $\gamma(S,\theta)$ density variables are the holy grail.

**Please specify the salinity variable. There are many different types of salinity variables and they all have a different interpretation and meaning to different communities. If it is practical Salinity, then use Sp. However, when using WOCE data, one should provide the data in Conservative Temperature and Absolute Salinity variables (that should be an editorial decision from a couple of years ago). Please use correct symbols along with it. If it is "model salinity", please be specific about that.**
Our study assumes fixed seawater composition and therefore relies on the use of Reference Composition salinity $S_R$, which is one of the particular type of Absolute Salinity discussed in TEOS10. As far as we are aware, this has been the default assumption in the overwhelming majority of oceanographic studies since the introduction of the last thermodynamic standard EOS81. It would seem therefore that there is only a need to be more specific in cases one would want to account for variable composition seawater by means of Density Salinity, another type of Absolute Salinity, as recommended by TEOS10. Note that Practical salinity is just one particular way to express Reference Composition Salinity, in the same way that degrees Celsius, Kelvins or Fahrenheit

represent different ways to express in-situ temperature. In all cases, there exists a one-to-one linear relationship allowing one to be expressed in terms of the other. As a result, even though the software used in our study to compute density takes practical salinity $S_p$ as its argument, it could easily be converted with two lines of code to take reference salinity $S_R$ as its argument. This one-to-one correspondence between $S_p$ and $S_R$ makes it possible to regard all oceanographic studies as using $S_R$ as their salinity variable rather than $S_p$, contrary to what the referee seems to believe. As to the referee's remark that the use of Conservative Temperature and Absolute Salinity should be imposed by editorial decision, we believe that it would be a sad day for science and academic freedom if it ever were to happen, since it would be akin to a decision to ban the use of the Celsius and Fahrenheit scale in favor of the absolute scale to express in-situ temperature; we therefore can only hope that the referee's remark was made in jest.

**Please provide a simple graph that illustrates the angles and vectors discussed in the introduction, specifically regarding equation (1)-(3), 11 and 15.**
OK thanks for this idea

**I'am a bit confused about the use of a horizontal and vertical diffusion tensor (Eq. 1). Common practice is to define a isoneutral (mesoscale) and isotropic (small-scale) mixing tensor, not a horizontal and vertical one. Not for convenience, but because of the different physical processes that are related to it. This part should be rewritten to include an isotropic and isoneutral mixing tensor and relate the related physical processes. If not, it should be clearly explained here how the horizontal and vertical mixing coefficient are justified, related to the physical processes that allow for the definition of vertical and horizontal mixing instead of isoneutral and isotropic mixing coefficients.**
We are surprised by this comment because we make it clear in our paper that the

use of horizontal/vertical diffusion is something that was done in early numerical ocean models, not in the most recent ones. As is well known, one of the main problem with horizontal/vertical diffusion tensors is that they are affected by the so-called Veronis effect, which historically is why they were subsequently replaced by the use of neutral rotated diffusion tensors. The main aim of this part is to demonstrating the usefulness of the concept of effective diapycnal diffusivity, by showing that it is naturally capable of quantifying the Veronis effect.

**In addition to the above, Eq (9) diffuses along a density surface. Hence I disagree with the use of Kd in this equation. This is essential small-scale mixing, which is isentropic. This means that it does not require a rotation into the diapycnal direction and should not be in this equation. I refer to section 4 of McDougall et al 2014 for further reading. Please adapt this in Eq. 9 up until the discussion of this term below eq. 11. If this term is removed from Eq. (9), then Eq. (9) represent the gradient of $\gamma(T, S)$ along the actual isoneutral direction. Of this, the component across its own surface is taken, which then provide the diapycnal transport, due to dianeutral diffusion of itself.**

This point was also raised by Prof. McDougall in a separate comment. Our reply is repeated here. We do not understand this comment. Indeed, as far as we understand it, the standard rotated diffusion tensor $K = K_i(I - dd^T) + K_d dd^T$ describes isotropic mixing as it is, at least as far as the locally-reference potential density $\rho_{lr}$ is concerned, since the form of $K$ implies for the latter:

$$\frac{D\rho_{lr}}{Dt} = \nabla \cdot (K_d \nabla \rho_{lr}) + N.L., \tag{1}$$

where N.L. refers to the terms arising from cabelling and thermobaricity. As far as we are aware, small-scale mixing of potential temperature (or Conservative Temperature) and salinity ($S_R$ or $S_P$) is not isotropic, based on the study by Smith and Ferrari (2009). However, even if one were to accept the idea of isotropic mixing for $\theta$ and $S$, it would only mathematically amount to modify the above tensor as follows:

$K^* = (K_i + K_d)(I - dd^T) + K_d dd^T = K_i^*(I - dd^T) + K_d dd^T$, where $K_i^* = K_i + K_d$ is a modified isoneutral turbulent mixing coefficient. Clearly, the modified diffusion tensor has exactly the same structure as the non-modified one; moreover, since $K_i$ is about 7 orders of magnitude larger than $K_d$, with both coefficients having large uncertainties, $K_i^*$ and $K_i$ are clearly indistinguishable from each other.

**P3-L23-28: Defines isoneutral, isopycnal, dianeutral and diapycnal. I like that. However, it would be worth moving this to an earlier stage in the manuscript. On top of that, throughout the manuscript there are places where isoneutral, isopycnal, dianuetral and diapycnal are confused. For example,**

- **P3-L32: Second sentence: Isoneutral contribution to diapycnal? Should this not be isopycnal to dianeutral?**

- **P4-L26, Isoneutral or isopycnal?**

- **P4-L31, Again, isopycnal not isoneutral. According to your definition, isoneutral mixing is by definition along neutral surfaces and has no diapycnal component. Please read carefully through this manuscript and make sure this is corrected at every point.**

We believe that all example that you give are correct in our manuscript. For example "isoneutral contribution to diapycnal" means that we are calculating the contribution of the isoneutral mixing on the mixing across our $\gamma$ variable. In numerical models it is customary to define the mixing according to the neutral and isoneutral directions. The question that we address here is: using a $\gamma(S, T)$ variable as defined in the manuscript to assess the dia-$\gamma$ mixing ( or diapycnal according to our definition) what part of it is due to isoneutral mixing ?

**How exactly is the neutral direction calculated? Calculating the exact neutral direction (d) can be tedious. Details why so, can be found in Appendix B of Groeskamp et al (2016) and probably also in Griffies et al (1998), although I'm not 100% sure of that. This has not been clarified in this paper and I'm wondering how this is done, as this is a major part of the analyses.**

The neutral vector is calculated from the gradient of the locally referenced density and used to calculate the angle between the neutral vector and $\nabla \gamma$ in formula (A1)(we have added this precision in appendix A). As the neutral vector is not calculated from the gradient of $\gamma^n$ i.e. we don't use $s_x = - \left( \partial \gamma^n / \partial x \right) / \left( \partial \gamma^n / \partial z \right)$, we do not have to use an interpolation method as described in appendix B of Groeskamp et al 2016 to avoid spikes when $\partial \gamma^n / \partial z \approx 0$. Note that our results show that $\gamma^n$ is not everywhere exactly perpendicular to the neutral direction. As a result calculating the neutral vector with $\gamma^n$ might not be the best choice in some regions.

**1   Comments**

**P2-L24, reference McDougall for Helicity.**
ok, thank you

**P3-L34: I'am not sure if this statement is correct. If you choose to calculate WMT due to sigma2, but you know the neutral direction everywhere then I think you may be able to calculate the WMT due to non-neutrality by exploring the sigma2 gradient in the neutral direction. For interpretation of the WMT it is still better to have an "as neutral variable as possible", but I think the statement here is too strong.**
The idea behind this sentence is to say that the total diapycnal diffusion through a

surface cannot be attributed only to dianeutral diffusion since neutral surfaces do not exist (it might be small, we don't know yet at this stage, but it cannot be zero).

**P4-L13: Please explain "Binary Fluid".**
ok

**P4-L16: What density are you talking about here? Density depends on pressure to. Are you here talking about y(S,T)?**
ok, we have replaced density by density-like variable

**P4-L33: Neutral density is categorized as a surface of y(S,T), but it also depends on P, x and y.**
you are right, we have added a sentence to explain why we have made this choice

**P5-L10: This Tensor is defined in Redi(1982), but was not correct for the small slope approximation (GM90). This is explained in McDougall et al (2014). Another version of this tensor is given by Griffies 1998. Few references are at place here.**
ok but we don't use the small slope approximation.

**P5-L20 I'm perfectly happy if you do not include the nonlinear terms, specially because they are their own form of flux. But a little sentence arguing why these are excluded would be nice.**
yes, it sounds like a hazardous approximation to us to include the nonlinear terms in the effective diffusivity term since there is no reason to think that the NL behave like a diffusive one. Therefore we avoided doing so. Note that in our calculation we do not neglect or do any hypothesis on the non-linear terms. We have removed this sentence

so that it is now hopefully less confusing: the diffusive flux of $\gamma$ is just (obviously) $-\mathbf{K}\nabla\gamma$.

**P6-L20 I think it should be $V(y0, t)$.**
no sorry we don't think so.$V(y, t)$ is the volume of ALL water parcels of reference density y0 satisfying $ymin < y0 < y$

**P6-L20-25, this was a bit hard to understand. A simple schematic would help.**
ok

**In Fig 3., why is the black line smaller than all the others, at great depths?**
The black line is $\gamma^T$, it thus suggests that $\gamma^T$ is closer to the neutral vector than all other variables under consideration here at great reference depth. Note that it is not "great depths" but great reference depth which is not exactly the same (sorry if you already know that).

**P11-L35 - Maybe not Neutral density, if you remove the 5%.**
the 5% calculation is just made to show that the discrepancy are very localized in space, we are not suggesting to use the neutral density whitout the 5%

**In section 3, $10^{-4}$ is chosen as a sort of a reference diffusivity. This seems a bit random. First of all, $10^{-4}$, is actually very diffusive for an ocean, as in most places it is way lower. Recent inverse and observationally based methods have shown this is probably much smaller, for global values (Zika et al 2010, Groeskamp et al 2017, Waterhouse et al 2014, Lumpkin and Speer 2007, etc). Also, I'm wondering, does the error not depend on what the local rate of dissipation is? If the local rate of dissipation is $10^{-3}$ (near topography) and the**

**error is $10^{-4}$ from this method, does that not mean the error is still relatively small? I suggest a deeper discussion of the use of $10^{-4}$ and the relation to local diapycnal mixing.**

$10^{-4}$ refers to the widely cited Munk and Wunsch figure, which is widely regarded as a canonical value. We agree that, as it is, it is not very clear. We have therefore added the MW reference. First of all, one of the most important point is that the effective diapycnal(or dia$\gamma$) mixing depends on the $\gamma$ variable used so that $10^{-4}\ m^2/s$ could be considered as "large" for a density variable that is approximately neutral but small for a variable that is far from being neutral such as $\sigma_0$. Secondly, our figures are computed from global-mean calculation and are thus not made to be compared with local values. Concerning the variables with local gradient close to the neutral vector, the high values that we found are due to very localized discrepancies (localized on less than 5% of each surface) between the neutral vector and the $\gamma$ gradient this is why it is not in contradiction with the fact that global values give much smaller values than $10^{-4}$ at least for $\gamma^n$. Plus: inverse and observationally based method calculate local values of mixing in the neutral direction i.e. dianeutral mixing (it seems to be the case at least for Groeskamp et al 2017 and waterhouse 2014), we calculate the effective diapycnal mixing through a $\gamma(S,T)$ due to isoneutral mixing which is not the same .

**P13-L5: This discussion should include a reference to the attempt of McDougall and Jacket 2005 to quantify the terms in the material derivative of Neutral Density, as some of the terms are small. What does that mean for Neutrality?**

Thanks for suggesting the reference to McDougall and Jackett (2005), which will be done in the revised manuscript. Regarding the consequences for neutrality, we don't know yet. However, our study points to a clear path to elucidate the issue. Indeed, our study makes a prediction for the effective diapycnal diffusivity of a few density variables assuming that neutral rotated diffusion is correct. Because the density variables considered are mathematically well-defined, it should be possible, at least in principle, to make a prediction for their effective diapycnal diffusivity directly from first principles

(from the analysis of the non-averaged Navier-Stokes equations) by linking it to microstructure measurements. Provided that this is possible, we would have an objective and independent way to predict the effective diapycnal diffusivity, thus allowing us to test whether neutral rotated diffusion is the right way to mix heat and salt in numerical ocean models or not. This is an avenue of research that we are currently exploring and that we hope to report on in the near future, but which we are stiff far from having resolved yet.

**P13-L24: Which exact developments of this paper need to be taken into account and for what purposes exactly? This paper provides no information on how to reduce mixing by changing advection schemes. Please specify.**

As explained in our paper, the concept of effective diapycnal diffusivity is at the heart of most inverse methods as well as of some recent attempts at diagnosing spurious numerical mixing due to numerical advection schemes, but so far, the assumption has been that the contamination due to isoneutral mixing was small and therefore negligible. Our results are important for demonstrating that this assumption is not justified in practice, and hence that accurate ways to remove the contribution of isoneutral mixing in the above said methods will need to be figured out.

---

## Referee Comment (RC2) · S. Groeskamp (Referee) · 19 Dec 2017

Response to comment on the review.
Sjoerd Groeskamp, 18-12-2017.

Dear Hochet et al, thanks for your comments on my review.

Before I provide a few specific and final comments, I need to address certain aspects of the comments to the review.

In general, reviewers are professionals and arguably experts in the field. In addition, they spent several hours (or more) to understand your manuscript and provide a review that can be used to improve the accuracy, presentation and impact of the reviewed work. Of course, I do not expect authors to fully agree with every review, but I do at the very least expect authors to appreciate the attempt to provide a review that improves the accuracy, presentation and impact of the presented work.

The response to the review of Hochet et al., is focused almost entirely to argue against the review comments. In addition, the comments are at some point even called "a jest". This is unprofessional and will not benefit the review process, nor the improvements of the manuscript.

Even when the authors disagree with the review comments, authors should realize that if the reviewer(s) do not fully understand their manuscript after careful reading, many others will not either. Time would be better spent by providing a more carefully and clearer written manuscript, then only arguing against the review comments.

With that said, below are my final comments that explain my statement above and provide some more scientific questions.

**About General comment number 1 (around C2)**
First of all, perhaps I choose the wording poorly, i.e. "oversell" and "holy grail" and should have phrased my response more appropriately. I apologies. And indeed, I agree finding a materially conserved and as neutral as possible variable, is worthwhile.

This comment meant to address the issue that a clearer statement and discussion could be provided on the concept that variables are 1) either materially conserved, but neutrality optimized (and thus not exact), or 2) variables can be Neutral, but conservation is optimized (and not exact). Perhaps explain how this affects the interpretation of the results for the different products, and discuss why it is not said that one is better than the other, but it is simply a different tradeoff.

**About the second general comment, on salinity.**
The reviewer does not provide time-consuming professional reviews of scientific work, "for a jest".

First of all, I would like to refer the authors to the Editorial note from the Journal of Physical (Spall et al, 2013). Admittedly, I could not find a comparable note from Ocean Sciences, but it may exist as it is available for many oceanographic journals. This note urges oceanographers

to use the TEOS-10 variables for observationally based work. Hence, my comment and the editorial notes was serious, "not a jest" as suggested by the authors.

Secondly, the whole and simple point of this comments is only to specify the salinity that is used. Currently it is specified as (last line, page 1):
… and S the (practical) salinity, …
Hence, it is not clear which salinity is used. To reword it in terms of temperature as in the authors their response, it is not clear if you used Fahrenheit, Celsius or Kelvin. I don't mind which one you use, as long as it is clear and well-motivated and with the associated notation ($S_p$ if it is practical salinity).

**General comment at the bottom of C4 about KH and KV.**
The authors state that it has been 'clear' that horizontal/vertical diffusion was applied in old models. The reviewer did not experience this to be so. Perhaps explain more clearly why one would go back to "old model" way of describing a diffusion tensor with Kh and Kv, while progress has been made by using a rotated tensor? Perhaps consider using Eq. (4) of Klocker et al (2009, A new method for forming approximately neutral surfaces) which provides a similar measure, but using isopycnal diffusion. If the authors have convincing arguments to stay with their current method, then that is fine, but that would still warrant a discussion between the two approaches and a motivation why to use the final chosen method.

**The comment of C5, that follows from the previous comment.**
I think that K* in the comment described the Redi-tensor, which is different than using the isotropic version. That was exactly the point of the review comment, as described on page 2173 of McDougall et al 2014.

**Comment C6**
I now agree that I indeed pointed out things related to Isoneutral/isopycnal, etc. that were in fact already correct in the manuscript.

**Comment around C9 about the Munk value.**
The Munk number may be canonical, but we now know that that number is a combination of mixing at hotspots, and a smaller global background-type mixing. The latter is much smaller, while the former is much larger. So, it may be worth spending a few words on how much it means to compare this to $10^{-4}$, knowing there are large spatial variations that may change the interpretation of the results. Also note that the Groeskamp et al (2017) estimates are global, while Waterhouse et al (2014) is a collection of local measurements.

Michael A. Spall and Karen Heywood and William Kessler and Eric Kunze and Parker MacCready and Jerome A. Smith and Kevin Speer and Mark E. Fernau.  EDITORIAL. Journal of Physical Oceanography, 43 (5), 2013, doi:10.1175/JPO-D-13-082.1

---

## Short Comment (SC3) · 21 Dec 2017

We thank the referee for engaging with us on various points raised in our response to his initial comments. Like all authors, we appreciate comments that contribute to improving readability and clarity of our work, but are understandably less appreciative of comments that we feel distract from the issues discussed.

**About TEOS-10 variables**

The referee wrote in his initial review *However, when using WOCE data, one should provide the data in Conservative Temperature and Absolute Salinity variables (that should be an editorial decision from a couple of years ago).* In our view, such a statement is unduly autocratic and encroaching on academic freedom, which is why we said in our response that we were hoping that the referee could not possibly be serious. To justify his comment, the referee mentions an editorial note from the Journal of Physical Oceanography (Spall et al., 2013). The said note, however, is merely an official announcement that AMS journals are making an exception to their former editorial rule of not capitalising names of variables with regard to Absolute Salinity and Conservative Temperature, starting with the paper by Graham and McDougall (2013). Contrary to what the referee's comment suggests, such a decision by no means makes it mandatory for oceanographers to use such variables. Historically, it is the first time that a colleague appears to interpret the term 'recommendation' as 'obligation'. In our opinion, the best practice is always to stay as close as possible to what is measured, which is certainly not the case of Conservative Temperature and Absolute Salinity.

**Salinity variables**

Our study — like all oceanographic studies for the past 40 years or so — is based on the use of reference composition salinity $S$, which forms the basis for both the old EOS-80 and TEOS-10 thermodynamic standards, as well as for the large majority of realistic numerical ocean model studies. We recently realised that practical salinity is the same as reference composition salinity, expressed in different units. Since it is not usual to use different notations $T_K$, $T_C$ or $T_F$ to refer to in-situ temperature $T$ expressed in different temperature units (Kelvins, Celsius or Fahrenheit), we believe that it is similarly not justified to use the notation $S_p$ for practical salinity, because it

erroneously suggests that it is a different salinity variable than reference composition salinity variable, when this is clearly not the case. These points will be clarified in our revised manuscript. We thank the referee for bringing up the issue, and helping to clarify what has been an immense source of confusion so far.

**Density variables (About General comment number 1 (around C2))**

We do not understand what the referee is after. The materially conserved variables whose effective diapycnal diffusivity is estimated in our paper are 'naturally' existing variables that have not been artificially designed to satisfy materiality over neutrality. We just investigate the properties of such variables, and do not advocate using one particular density variable over another. Our results have no implications for determining whether one type of density variable might be better than another one, so we do not understand why the referee is asking us to discuss such implications.

**General comment at the bottom of C4 about $K_H$ and $K_V$**

We find it impossible to relate the referee's comment to what we do or write in our paper. What we do in our paper is to discuss the concept of effective diapycnal diffusivity $K_{\text{eff}}$. $K_{\text{eff}}$ is a quantity that can be defined for any material density variable $\gamma(S, \theta)$ for any kind of diffusion tensor $K$. What we do is to illustrate the concept for both the cases where $K = K_H(I - kk^T) + K_V kk^T$ mixes separately in the horizontal and vertical directions, as was done in early numerical ocean models, as well as for the modern neutral rotated diffusion tensor $K = K_i(I - dd^T) + K_d dd^T$ used in current ocean models, using the form given in Griffies (2004)'s textbook. Here, $k$ is the unit vertical vector, and $d$ the normalised neutral vector. We do not understand where in our manuscript does the referee think that we advocate going back to using non-rotated diffusion tensors,

which is a complete misrepresentation of our study. The concept of effective diffusivity discussed in our paper can be traced back to the studies by Nakamura (1996, 178 citations) and Winters and d'Asaro (1996, 105 citations), which are widely regarded as the two studies formalising the concept rigorously. As far as we are aware, the study by Klocker et al. (2009) (8 citations, of which 5 are self-citations) is not known as a study concerned with the problem of estimating effective diffusivities, so that we do not understand why we would need to justify using a well-established approach preferentially to an irrelevant one.

**Comment about $K^*$**

We do not understand what the referee is trying to say, nor do we understand the paper by McDougall et al. (2014). The referee needs to spell out exactly what he means, and what McDougall et al. (2014) are supposed to have showed or done in plain english. Just referring to a page number is not helpful. It would also be helpful if the referee could provide the physical basis for why he thinks that mixing of potential temperature and salinity is isotropic at small scales, when Smith and Ferrari appear to show otherwise.

**Comment on the Munk value**

The physical reason(s) for how to explain Munk's canonical value $K_v = 10^{-4} m^2/s$ are irrelevant to our discussion. What matters to our discussion is that $10^{-4}$ is a value that is universally accepted as being about 1 order of magnitude 'too large' compared to values most commonly observed in the main thermocline, and is therefore a useful benchmark in that regard. The only justification for using this value is that it is one that

everyone in the field can recognise owing to its historical importance in the context of ocean mixing.

**References**

- Klocker, A., T. J. McDougall and D. R. Jackett, 2009: A new method for forming approximately neutral surfaces. Ocean Sciences, 5, 155-172. (8 citations, 5 self-citations).

- Nakamura, N. 1996: Two-dimensional mixing, edge formation, and permeability diagnosed in an area coordinate. J. Atmos. Sci., 53, 1524-1537. (178 citations)

- Winters, K.B. and E. d'Asaro, 1996: Diascalar flux and the rate of fluid mixing. J. Fluid Mech., 317, 179-193. (105 citations)

---

## Short Comment (SC4) · 22 Dec 2017

It is not true that potential temperature is equally good as Conservative Temperature in an ocean modelling context. This has been described in several papers that quantify the non-conservative production of both variables, and I note that section 5 of the quoted paper of Tailleux (2015) is incorrect in this regard. Here is another way of seeing this. Consider the ocean surface which is exchanging heat with the overlying atmosphere continuously. When integrating over say a minute, or an hour or a day or a week or a season, when using Conservative Temperature it is simple to calculate the ingest of Conservative Temperature over this period of time; it is simply the ingested

amount of heat divided by a fixed value of heat capacity namely cp_0. But the ingested amount of potential temperature is impossible to calculate accurately when potential temperature is used as the model variable. This is because the ingested amount of potential temperature is equal to the time integral of the instantaneous air-sea heat flux divided by the specific heat capacity at zero pressure, cp(SA,T,0). This specific heat capacity varies each second and is different in the daytime versus at night time (because of the variations in temperature), and it varies with the sea surface salinity. These temporal variations of temperature and salinity are unknown to an ocean model at time scales less than the time step of the model. Hence it is impossible to accurately calculate the air-sea flux in terms of potential temperature. Similarly, in the interior of the ocean, due to unresolved temporal and spatial variations of the turbulent fluxes, it is not possible to accurately evaluate the non-conservative production terms. This is the reason why their magnitude should be made as small as possible, for example, by using Conservative Temperature rather than potential temperature as the model prognostic variable. Hence it is not true (as the present OSD paper and also Tailleux (2015) suggest) that potential temperature is equally good as Conservative Temperature as a choice for a prognostic variable for an ocean model.

Regarding the isotropic nature of small-scale turbulent mixing, both the Smith and Ferrari (2009) paper and the Scotti (2015) paper [Scotti, A., 2015: Biases in Thorpe-scale estimates of turbulence dissipation. Part II: Arguments and turbulence simulations. J. Phys. Oceanogr., 45, 2522-2543] support the isotropic nature of small-scale turbulent mixing. I agree that the correct treatment of small-scale mixing processes by allowing them to not only diffuse diapycnally but also epineutrally does cause only trivial changes to the amount of epineutral diffusion. Nevertheless, it is good to include these processes correctly so as to reduce confusion in the literature, and to use language correctly.

---

## Short Comment (SC5) · 22 Dec 2017

**Potential temperature versus Conservative Temperature**

McDougall (2003) pioneered the idea that 'non-conservativeness' is central to understanding how to ascertain the usefulness and accuracy of various heat variables for defining heat content and heat transport. His pointing out that potential temperature $\theta$ is much less conservative than previously assumed is a significant result with important implications. Historically, however, McDougall (2003) suggested two solutions to the problem. 1) Either make the existing equation for potential temperature more

accurate by adding the neglected non-conservative terms, or 2) replace the equation for potential temperature by one based on a more conservative heat variable. Prof. McDougall subsequently devoted a large part of his career promoting solution 2, centred on the use of potential enthalpy and Conservative Temperature, whereas I spent a large fraction of the last 10 years or so working on solution 1, centred on understanding how one might diagnose the non-conservation of potential temperature in order to make the potential temperature equation used in numerical ocean models as accurate as feasible. My main result was to show in Tailleux (2010) that the non-conservation of $\theta$ is what is needed to render total energy conservative, which Tailleux (2015) exploited to improve the accuracy of the equation for $\theta$, as well as for improving total energy conservation in numerical ocean models.

It seems important to point out that although the introduction of potential enthalpy by McDougall (2003) was an important stepping stone towards understanding ocean heat and energy conservation, it was clear from day one that it could not be the ultimate and definitive solution to the problem of how to define heat in the ocean. Indeed, using potential enthalpy as our definition of 'heat' requires defining dynamic enthalpy as our definition of 'work', which is inconsistent with Lorenz theory of available potential energy. Moreover, potential enthalpy and Conservative Temperature are still significantly non-conservative, especially in the deep ocean, so that it has always been clear from day one that a heat variable more conservative than Conservative Temperature should exist. As it turns out, such a variable — which is significantly more conservative than Conservative Temperature — has finally been found, and a paper discussing it will be submitted in the next few months. Unsurprisingly for those familiar with the issues involved, such a variable is closely related to the background potential energy entering Lorenz theory of available potential energy.

Prof. McDougall seems to have convinced himself that the temperature variable to be used in numerical ocean models should be the same as the variable used to define heat content and heat transport. But this does not need to be the case. Indeed, it

is perfectly possible to use potential temperature as many numerical ocean models still do, improve its accuracy by retaining its non-conservation as proposed in Tailleux (2015), and diagnose heat content and heat transport in terms of whatever quantity we think defines heat content best. In fact, this is the approach adopted in numerical atmospheric climate models, where there is no direct link with the quantity used to define heat (based on moist enthalpy for instance) and the temperature variable used (potential temperature or liquid potential temperature for instance). The main problem with formulating a numerical ocean model (or the Gibbs Seawater library) in terms of Conservative Temperature is that such a model or software library will become obsolete as soon as a more conservative heat variable is found, that is in 2018.

There is of course nothing wrong with Tailleux (2015), which Prof. McDougall recommended for publication at the time. I can only hope that badmouthing my paper will encourage — rather than put off – readers to decide for themselves what Tailleux (2015) is really about

**Isotropic mixing of potential temperature and salinity**

We are happy to stand corrected on the issue, and of course support the elimination of confusion about the nature of small scale mixing, but it would be more helpful if Prof. McDougall could explain in simple physical terms to the readers of Ocean Science and to us how having significantly different turbulent spectra for density-compensated $\theta/S$ anomalies versus non-compensated $\theta$/S anomalies, which is Smith and Ferrari's finding, is consistent with the idea of isotropic mixing for potential temperature and salinity. Would Prof. McDougall care to explain how does he define 'isotropic' mixing in terms of turbulence spectra?

**References**

- McDougall, T.J., 2003. Potential enthalpy: a conservative oceanic variable for evaluating heat content and heat fluxes. J. Phys. Oceanogr., 33, 945–963.

- Tailleux, R., 2010: Identifying and quantifying nonconservative energy production/destruction in hydrostatic Boussinesq primitive equation models. Ocean Modell., 34, 124–136.

- Tailleux, R., 2015: Observational and energetics constraints on the non-conservation of potential/ Conservative Temperature and implications for ocean modelling. Ocean Modell., 88, 26–37.

---

## Referee Comment (RC3) · Anonymous Referee #2 · 1 Feb 2018

Summary of key results

The paper estimates the effective diapycnal diffusivity due to "leakage" from the explicit isopycnal diffusion that arises from the local misalignment of five density variables commonly used in numerical ocean models and in analysis of hydrographic data from the neutral direction, namely the neutral density gamma_n of Jackett and McDougall (1997); the Lorenz reference state density rho_ref of Winters and D'Asaro (1996); and the potential densities referred to 0, 2,000 and 4,000 dbar pressure.

Using temperature and salinity fields from the WOCE climatological dataset, the authors calculate the angle between the gradient of the density fields and the neutral

direction and from this derive global mean profiles of an effective diffusivity Keff due to the projection of the (much larger) isoneutral mixing coefficient onto the dianeutral direction. For most of the density coordinates used, this angle is generally around 10-4 or less north of 40°S, and larger in the Southern Ocean; but in the case of the potential density sigma_0 it has larger values over much of the ocean interior. Integrated over all density surfaces outside the Arctic Ocean, and using two plausible choices of the isoneutral mixing coefficient Ai, this gives mean Keff profiles significantly larger than estimates made from hydrographic measurements, with the neutral density lowest and sigma_0 highest. However, if the 5% of points with the largest angles (mainly in the Southern Ocean) are removed from the calculation the effective diffusivities are substantially reduced, lying generally below 10-5 m2/s, apart from in the case of sigma_0.

Recommendation

The exposition is generally clear, logical and correct; the Introduction and Method sections being relatively straightforward to follow.

To my knowledge this is the first time such a study has been made, and I believe the results are of interest to both the observational and the modelling communities. If the changes I suggest here are incorporated I would be happy for the paper to be published.

General style

The paper is generally well structured and overall well written, although there are sections which appear to have been written by different co-authors, and the quality of English is uneven (for instance in the Abstract and the Conclusions).

The line numbering starts afresh each page, which makes referring to lines slightly more awkward for this Reviewer.

General comments

I think the results presented in the paper are interesting and novel, but it isn't clear to

me exactly what the practical recommendations of the study are. You have shown that the non-zero angle between the gradient of the density variable and the direction of diffusion give significant leakage from isoneutral to dianeutral mixing, which is generally smaller with the neutral and reference densities than with the potential densities, and you make the point that the neutral density is less useful in this context since it is non-material. You demonstrate that this leakage creates global mean effective mixing coefficients substantially larger than 10-4 m2/s below 3,000m depth, but when the largest 5% of the angles are removed from the sum the global mean Keff is reduced by an order of magnitude. I would have thought that the latter is the more interesting and useful number, since it is more typical of the ocean outside the Southern Ocean. In addition, when this pesky 5% are removed the profiles for the different density variables also get much closer together, implying that the choice of density variables has a stronger influence on the extreme values of the angle than on the more typical regime in the other 95% of the ocean. Although the discussion in the Conclusions section is relevant and valid, I feel that the qualifications above should be included in the overall conclusions.

I have my doubts about Figure 3, as I detail below.

Specific corrections:

Abstract

P1L3: Replace "impossibility to construct" with "impossibility of constructing" P1L9: Replace "isoneutral mixing" with "the isoneutral mixing coefficient" P1L10: Replace "yields values systematically" with "yield values consistently" P1L14: Replace "masses" with "mass" and insert "a" before "Lorentz".

1. Introduction

P1L18: Need to use either "sub-grid scale" or "subgridscale" consistently throughout – not both! P1L23: This is the second occurrence of "indeed" in this paragraph: it

reads clumsily. P3L7: It would help the clarity of this section considerably if a sentence or so summarising what McDougall and Jackett mean by "fictitious mixing" were included here, as well as a clear statement of how it differs from the effective mixing discussed here. This whole paragraph, in my opinion, is too long and too sprawling in structure. I would suggest it is restructured more logically and clearly into two or three separate paragraphs. P3L20: It would be appropriate to mention here that Megann (Ocean Modelling, 2018) recently showed that the Lee et al approach gave diapycnal transformation rates in a $\frac{1}{4}°$ÂăNEMO model that were not especially sensitive to the choice of potential density coordinate used. P4L10: "There is no question that..."; I would dispute that, since the APE method has generally only been used for a model that is unforced (spinning down, that is), so does not give the complete picture of the numerical mixing that occurs when the model is run in a more "normal" and useful way. P4L19: This section would be clearer if the Lorenz reference density were defined earlier in the paragraph, so that its relevance to the Lorenz reference state were more obvious. P4L23: "detph"? P4L32: Add "neutral density" before "gamma-n".

2.1 Effective diffusivity

P5L9: The equations for theta and S evolution do not include forcing terms, and this needs to be stated explicitly here (which is stated later on for Equation 17). P5L14: Add "as" after "given".

2.2 Reference profile

P6L25: insert "the" before "Lorenz". P6L17: I think it would be worth stating that this definition of zr is only strictly valid where the selected density coordinate is monotonic everywhere with depth (which is not the case, for instance, with the potential density coordinates). P7L18: Insert a semicolon after Equation 18, and "this" at the start of the line.

3 Isoneutrally-controlled effective diapycnal diffusivities

P8L4: Replace "calculation" with "calculations" P8L5: It is not immediately clear why gamma-n. is not defined north of 60°N (at least in a way that the other density variables are). P9L3: Replace "sinus" with "sine". Figure 3: Is there a mistake here? Panels A and B appear to be identical, where I would expect the values in B to be quite a bit different, since Ki is a multiplier in the expression for Keff., and presumably the former is quantitatively rather different in the two cases? P10L1-2: Replace both occurrences of "amount" with "number". P10L2: Replace "overcome" with "dominate". P11L7: If lines 7-11 were incorporated into the previous paragraph and a new paragraph break inserted before Line 12 the structure would be easier to follow. P11L15: Again, I am not convinced that the results for cases A and B can really be so similar, since the value chosen for the constant Ki=1,000m2s-1 used in case A is essentially arbitrary, and it would be an extraordinary coincidence if it produced results so similar to those in case B. Figure 4: The colour legend would be easier to interpret if the annotations of the log scale were in integer increments, rather than the apparently uneven ones (approximately, but not exactly, 0.9!) used here.

**4 Conclusions**

The discussion is relevant and interesting, but the conclusions need to be clarified, as I suggest above. As I mentioned earlier, the analysis that flows from Equation 7 is only strictly valid in the absence of surface forcing. It should therefore be noted, particularly in the discussion of Figure 4, that much of the Southern Ocean - as well as the Atlantic north of 50°N - is directly ventilated and so a good argument could be made for excluding it from the global mean in this calculation. I would guess that this might be a physically-based argument for the exclusion of the 5% of points that have large angles; perhaps coincidentally corresponding roughly to the directly ventilated regions. It would also be informative if the profiles obtained for Keff using the various density definitions were at least qualitatively compared and contrasted in this section with those estimated from observations, with those used in model mixing schemes, and also with those diagnosed for numerical mixing in models by the studies already cited here (which can

be an order of magnitude larger than the former). This comparison would put the calculated Keff values in context, and would also illuminate the importance (or not) of the 5% of extreme values for the angles in the global means.
* * *

---

## Author Comment (AC2) · 16 Mar 2018

**Final Response**

**Antoine Hochet and Rémi Tailleux**

**March 16, 2018**

We thank Prof. McDougall, Dr. Groeskamp and anonymous Referee 2 for their generally positive assessment of our paper and for their comments/suggestions for clarifying it. In this final response, we synthesize the outcome of the discussion/exchange of views and explain how we plan on revising the paper to address these. This is done point by point for the comments by anonymous Referee 2, as these are addressed here for the first time, but thematically for the comments raised by Prof. McDougall and Dr. Groeskamp, since these have already answered point by point in previous responses published on the discussion page.

**1 Response to particular points**

**1.1 Isotropic character of the dianeutral component of rotated diffusion tensors**

Both Prof. McDougall and Dr. Groeskamp have suggested to make the dianeutral component of rotated diffusion tensors isotropic as per a recommendation by McDougall et al. (2004). In a previous response, we counter-argued that as far as we understood the issue, whether to make the dianeutral component of the rotated diffusion tensor isotropic or not depended on one's definition of the isoneutral mixing coefficient $K_i$. In our paper, we adopted the formulation of rotated diffusion given by Eq.(7.9) of Stephen Griffies textbook 'Fundamentals of Ocean Climate models', which we believe is the one currently in use in the majority of current numerical ocean climate models. As explained in a previous response, McDougall et al. (2014)'s suggestion is physically equivalent to replace our definition of $K_i$ by $K_i + K_d$; since $K_i$ is about 7 orders of magnitude larger than $K_d$, McDougall et al. (2014)'s suggestion amounts to argue that it is somehow possible to know the normalized value of $K_i$ up to its $7th$ decimal, which is clearly not true given current uncertainties on mixing measurements. In a previous response, we also expressed our surprise at the view that the dianeutral component of rotated diffusion should be made isotropic, given that it seems to conflict with our interpretation of the results by Shafer-Smith and Ferrari (2009). Indeed, in this paper as well as in others, turbulence spectra are usually found to be very different in the dianeutral and isoneutral directions, which in our view is the definition of anisotropy. It is to be noted that McDougall et al. (2014) appear to regard their suggestion as self-evident, as they do not provide any reference in support of their view. *Action: Discussion of the above points will be added in the revised version of the manuscript.*

**1.2 Definition of salinity. Potential temperature versus Conservative Temperature**

We will clarify our definition of salinity in the revised manuscript along the lines mentioned in our previous response to comments. Regarding the use of Conservative Temperature, we accept Prof. McDougall's result that CT is indeed more conservative than potential temperature. While it would be relatively easy to redo the computations with CT, this is not needed here because the concept of effective diffusivity does not require our temperature variable to be as conservative as possible. This is because the non-conservation of potential temperature only enters the expression for the nonlinear production of the density variable considered, not the effective diffusivity.

**1.3 Other comments**

All remaining comments and points raised by Dr. Groeskamp and Prof. McDougall will be addressed in the revised manuscript along the lines discussed in our responses to comments already posted. One important application

of our results is in the construction of a simplified representation of ocean heat uptake for use in simple climate models, as explained in the preprint posted on https://arxiv.org/abs/1708.02085

**2   Comments raised by Anonymous Referee 2**

We thank the referee for his/her comments and for being generally supportive of our paper. We are also very grateful for his/her many correction of our English that will be included in the revised manuscript. Our response to his/her comments are provided below.

**General Comments**

*I think the results presented in the paper are interesting and novel, but it isn't clear to me exactly what the practical recommendations of the study are. You have shown that the non-zero angle between the gradient of the density variable and the direction of diffusion give significant leakage from isoneutral to dianeutral mixing, which is generally smaller with the neutral and reference densities than with the potential densities, and you make the point that the neutral density is less useful in this context since it is non-material. You demonstrate that this leakage creates global mean effective mixing coefficients substantially larger than $10^{-4}$ m$^2$/s below 3,000m depth, but when the largest 5% of the angles are removed from the sum the global mean Keff is reduced by an order of magnitude. I would have thought that the latter is the more interesting and useful number, since it is more typical of the ocean outside the Southern Ocean. In addition, when this pesky 5% are removed the profiles for the different density variables also get much closer together, implying that the choice of density variables has a stronger influence on the extreme values of the angle than on the more typical regime in the other 95% of the ocean. Although the discussion in the Conclusions section is relevant and valid, I feel that the qualifications above should be included in the overall conclusions*

okay, thank you, we will discuss further on the 5% in the conclusion.

**Specific corrections**

- *It would help the clarity of this section considerably if a sentence or so summarising what McDougall and Jackett mean by "fictitious mixing" were included here, as well as a clear statement of how it differs from the effective mixing discussed here.*
  The distinction between McDougall and Jackett's concept of fictitious mixing and our concept of effective mixing is discussed in details in the introduction of our paper. Perhaps the referee missed this?

- *It would be appropriate to mention here that Megann (Ocean Modelling, 2018) recently showed that the Lee et al approach gave diapycnal transformation rates in a 1°NEMO model that were not especially sensitive to the choice of potential density coordinate used.*
  Thank you for pointing out the relevant reference to Megann (2018). The lack of sensitivity to the choice of potential density is interesting, for it suggests that the effective diffusivity computed by Megann (2018) might then be dominated by spurious numerical mixing. We will add a comment to this point in our revised manuscript.

- *P4L19: This section would be clearer if the Lorenz reference density were defined earlier in the paragraph, so that its relevance to the Lorenz reference state were more obvious.*
  Thank you for the suggestion, which we will implement in our revised manuscript.

- *P4L10: "There is no question that. . ."; I would dispute that, since the APE method has generally only been used for a model that is unforced (spinning down, that is), so does not give the complete picture of the numerical mixing that occurs when the model is run in a more "normal" and useful way.*

Note here that we were careful in our paper to restrict our statement to the case of a linear equation of state. Even though we agree that Winters et al. 1996 rigorous approach to quantifying diapycnal mixing has been primarily restricted to freely decaying simulations, there is in principle no problems with applying the method in a forced-dissipated context as shown by the following formula:

$$\frac{\partial \gamma_r}{\partial t} = \frac{1}{A(z_r)} \frac{\partial}{\partial z_r} \left( A(z_r) K_{\text{eff}}(z_r) \frac{\partial \gamma_r}{\partial z_r} \right) + \text{forcing} \tag{1}$$

with:

$$\text{forcing} = -\frac{1}{A(z_r)} \frac{\partial}{\partial z_r} \left( \int_{V(z_r)} f_\gamma dV \right) \tag{2}$$

The derivation of this equation is shown in appendix B. So if $f_\gamma$ is known for each time step, it can be integrated in each $z_r$ classes and then used in equation (2) and (1) to obtain $K_{\text{eff}}$. Thus in theory unforced spinning down experiments are not required to use this method. The method is considerably more difficult and subtle to use for a nonlinear equation of state, however, because then cabelling and thermobaricity introduce an additional diapycnal flux that is hard to estimate and not easily separated from the 'linear equation of state' part of the diapycnal diffusive flux. In any case, these considerations are not directly relevant to our paper, because our approach to estimating effective diapycnal mixing is not based on monitoring the evolution of the reference state, but on numerically estimating the analytical expression obtained for it. In this respect, our approach is very similar to that used by Jackett and McDougall for estimating fictitious mixing. APE theory only enters the problem here for the purpose of introducing Lorenz reference density as a generalised form of potential density that is more neutral than any other existing potential density. In contrast to Winters et al. approach, we do not use sorting at all.

- *Figure 3: Is there a mistake here? Panels A and B appear to be identical, where I would expect the values in B to be quite a bit different, since Ki is a multiplier in the expression for Keff., and presumably the former is quantitatively rather different in the two cases?*
  We have checked that there is no mistake. The results in the two panels are indeed very similar but not identical (which is undeniably hard to see from panels A and B). The values on panels A and B are mostly set by only 5% of the points on each surface and it appears that for this 5% (located mostly in the ACC) the isoneutral mixing coefficient is of the order of 1000 m$^2$/s which leads to very similar $K_{\text{eff}}$ curves when displayed on a log scale. We did the same calculation for $\sigma_2$ without the 5% of the largest values on each $z_r$ surface and obtained much more contrasted $K_{\text{eff}}$ between the case with $K_i = 1000 m^2/s$ and $K_i$ of Forget et al. 2015.
  We will add a sentence in the manuscript to explain why their is this similarity between panel A and B.

- *Figure 4: The colour legend would be easier to interpret if the annotations of the log scale were in integer increments, rather than the apparently uneven ones (approximately, but not exactly, 0.9!) used here.*
  okay, we will improve this.

- *As I mentioned earlier, the analysis that flows from Equation 7 is only strictly valid in the absence of surface forcing. It should therefore be noted, particularly in the discussion of Figure 4, that much of the Southern Ocean - as well as the Atlantic north of 50N - is directly ventilated and so a good argument could be made for excluding it from the global mean in this calculation. I would guess that this might be a physically-based argument for the exclusion of the 5% of points that have large angles; perhaps coincidentally corresponding roughly to the directly ventilated regions.*
  We disagree with the referee's interpretation of our approach, which, as stressed above, is very different in its principle from Winters et al.'s sorting approach. We insist that our analytical expression for the effective diffusivity is valid for a forced/dissipated ocean with a nonlinear equation of state, as is the case of Jackett and McDougall's expression for the fictitious diffusivity.

- *It would also be informative if the profiles obtained for $K_{eff}$ using the various density definitions were at least qualitatively compared and contrasted in this section with those estimated from observations, with those used in model mixing schemes, and also with those diagnosed for numerical mixing in models by the studies*

*already cited here (which can be an order of magnitude larger than the former). This comparison would put the calculated Keff values in context, and would also illuminate the importance (or not) of the 5% of extreme values for the angles in the global means.*

Ok thank you for this idea. We will add in the revised manuscript a figure showing the same $K_{eff}$ but calculated from an estimation of local dianeutral diffusivities (from the same Forget et al. 2015 database) to show how it compares with the one obtained from the isoneutral diffusivities and discuss qualitatively the values with those found in the literature.

---

## Referee Report (RR1)

Review of
**Isoneutral control of effective diapycnal mixing in numerical ocean models with neutral rotated diffusion tensors**
[Hochet, Tailleux, Ferreira, Kuhlbrodt]

One cannot define a density variable that is both a material surface, and along which parcel exchanges are energetically neutral.  Thus ocean models that rotate the diffusion tensor (a la Redi 82) to a neutral frame, in order to separate large isopycnal from much smaller diapycnal diffusivities, will incur inadvertent diapycnal diffusion due to the isopycnal part. This paper seeks to make this clear, and compare the relative spurious mixing errors incurred by the use of various neutral density variables.  The analysis is done not with an ocean model, however, but with hydrography.  The results are that it doesn't matter too much, except in about 5% of the locations.

My chief concern is that, as implied above, while this paper seems to be directed toward ocean models, no attempt is made to connect their results to what is actually done in models, beyond the idea of the rotated Redi diffusion tensor. What neutral vectors do MOM, NEMO and the MITgcm use, for example?  How would the present results depend on model resolution?  Are the implied errors significant relative to other errors, or are they negligible?  I very sincerely have no idea what to do with the information the authors have given me, or whether it is something a real modeler needs to worry about, or not.

This confusion allows for some lofty and silly claims in the conclusion, especially the last two sentences of the paper, where physical ideas are conflated with a modeling issue.

**Specific points**

*P 3, L18-28:* This argument is indecipherable.  What is \hat\rho? Where does (7) come from? (the authors say "The expression for the neutral vector *becomes*…" 'becomes' from what?  And there is an error below (8): that J must be \partial\gamma/\partial S, not the Jacobian term, which is already in (8).

For reference, I have read the relevant parts of Tailleux (2016b), which is well-written and careful.  In the present paper, the authors are sloppy and assume too much of the reader.

*P 5, L 17:* Reference Gent and McWilliams 90 and at least briefly explain v_gm!

*Sec. 2.2 & Appendix B:* Conversion to density (or any other tracer) coordinates has been done over and over in the literature.  Admittedly, it's a messy literature, but it doesn't need to be repeated here. In particular, there is only one effective diffusivity.  I haven't read Speer 97, but I've read WD96, Nakamura 96 and Shuckburgh and Haynes 03 (SH03) in great detail — the latter is the most useful.  There is an error in the effective diffusivity formulation of WD96 that stems from their incorrect eq. (9), and it

seems to have made its way to this paper in (18), though I am not certain because I'm a little uncertain about \nabla z_r.   I would in any case recommend the authors look at SH03, equations (6) & (7) (and compare to WD96 eq. (12)).  That said, I doubt it will affect the results much.

Regarding Speer's Keff, is (20) correct?  It would say Keff = K if K is constant… surely that's not right.

*Sec 3*:  I am very surprised that none of the reviewers asked for this:  please add a detailed description of the algorithm used to compute your Keff.  Given the errors and mess in the literature, it's essential.

Throughout: There are misspellings and incorrect uses of plurals and s endings.  e.g. "…the small number of pointS with …" and "… could correlates …" and "This is at oddS with …"

---

## Author Response (AR2)

**Answer to anonymous referee #3**

**September 14, 2018**

We thank the referee for his/her comments and requests for further clarification of various parts of our paper, especially its main aims. In response, we have rewritten most of the abstract, and expanded some of our arguments in the unclear parts of the paper indicated by the referee. We have also removed most of the mathematical contents from the introduction to put it in subsequent sections or in the Appendix to facilitate understanding of the main aim and objectives of the paper. We hope that the referee will find our response satisfactory and the revised paper sufficiently improved to warrant publication. Note that the parts highlighted in blue in the revised version of the paper are the ones corresponding to new material not present in the previous version of the paper. Material that has been simply shuffled around or simply rephrased in only minor ways for clarity has not been highlighted in blue.

**'One cannot define a density variable that is both a material surface, and along which parcel exchanges are energetically neutral. Thus ocean models that rotate the diffusion tensor (a la Redi 82) to a neutral frame, in order to separate large isopycnal from much smaller diapycnal diffusivities, will incur inadvertent diapycnal diffusion due to the isopycnal part. This paper seeks to make this clear, and compare the relative spurious mixing errors incurred by the use of various neutral density variables. The analysis is done not with an ocean model, however, but with hydrography. The results are that it doesn't matter too much, except in about 5% of the locations.'**

**Response:**
We disagree that our paper is about comparing the relative spurious mixing errors incurred by the use of various neutral density variables. Indeed, the idea that spurious mixing would result from constructing a rotated diffusion tensor based on mixing directions other than the isoneutral and dianeutral directions relies on two main assumptions: 1) the measured diapycnal values diapycnal values of $O(10^{-5}m^2/s)$ pertain to the neutral rotated diffusion tensor $\mathbf{K}_n$, not $\mathbf{K}_\gamma$; 2) that an ocean modeller would change the mixing directions without altering the values of the mixing coefficients. Physically, it is key to realise that each density variable $\gamma$ comes with its own distinct diapycnal diffusivity $K_\gamma$. As a result, changing the mixing directions in a rotated diffusion tensor without changing the mixing coefficients is what would cause spurious diapycnal mixing; the correct approach of changing both the mixing directions as well as the mixing coefficients in a consistent way would a priori not introduce spurious mixing. As a result, we believe that the contamination of diapycnal diffusivities $K_d^\gamma$ pertaining to density-like variable $\gamma(S,\theta)$ by isoneutral mixing should be a real physical effect if we accept that the dianeutral and isoneutral mixing coefficients entering neutral rotated diffusion tensors are the correct ones. The results are important because the diapycnal diffusivity $K_d^\gamma$ pertaining to a globally-defined density variable $\gamma$ is a priori easier to constrain from observations using Walin-type water mass analyses than $K_d^n$, even if it is the latter one wants to know about in practice. To that end, however, one needs to understand how the various diffusivities are inter-related. Rather than saying that the above issue does not matter too much except in about 5% of the locations, it seems more accurate to regard our results as stating that the problem matters whenever closeness to neutrality is hard to achieve, such as in the Southern Ocean for instance, which is a key region controlling many aspects of the large-scale ocean circulation. In other words, the potential importance of the problem is not necessarily well quantified by apparent smallness of the number 5%, although it is hard to say much more at this stage.

**Action**: We have rewritten the abstract and parts of the conclusion to make the above ideas clearer.

**My chief concern is that, as implied above, while this paper seems to be directed toward ocean models, no attempt is made to connect their results to what is actually done in models, beyond the idea of the rotated Redi diffusion tensor. What neutral vectors do MOM, NEMO and the MITgcm use, for example? How would the present results depend on model resolution? Are the implied errors significant relative to other errors, or are they negligible? I very sincerely have no idea what to do with the information the authors have given me, or whether it is something a real modeler needs to worry about, or not.**

**Response**: We agree that the paper is indeed directed toward ocean models, but not in the sense assumed by the referee and only to some extent. Indeed, the underlying question addressed by our paper is how can we use observations to constrain the dianeutral and isoneutral mixing coefficients entering neutral rotated diffusion tensors, given that such coefficients do not a priori relate to the mixing of any globally-defined density variable? In this regard, our paper should be of interest to ocean modellers, because it is concerned with the fundamentals of how to construct and constrain neutral rotated diffusion tensors. But our paper is also, perhaps more importantly, directed towards the oceanographer interested in constraining the dianeutral mixing by means of a Walin-type water masses analyses. However, because such an approach yields information about the diffusivity pertaining to the particular density variable $\gamma$ used to do the analysis, one needs to understand how to relate the $\gamma$-diapycnal diffusivity to the dianeutral and isoneutral mixing coefficients. The main point of the paper is that it is not really possible for Walin-type inverse methods to constrain the dianeutral diffusivity without knowing something about the isoneutral diffusivity. The same ideas pertain to the estimation of spurious diapycnal mixing by means of the Winters et al. approach, because such an approach can be viewed as a Walin-type water mass analysis based on the use of Lorenz reference density. Re-reading the manuscript, it seems to us that the referee's confusion might be due to the failure of the abstract to convey our message sufficiently clearly, so we decided to rewrite it entirely. Given that our paper is not directed toward ocean models, we feel that the questions of what neutral vectors are used by MOM, NEMO or the MITgcm are irrelevant to our argument as is model resolution. **Action**: We have rewritten parts of the abstract, introduction and conclusion to clarify the issues.

**Specific points**

**P 3, L18-28: This argument is indecipherable. What is $\hat{\rho}$? Where does (7) come from? (the authors say 'The expression for the neutral vector becomes...' 'becomes' from what? And there is an error below (8): that J must be $\partial\gamma/\partial S$, not the Jacobian term, which is already in (8).**
We believe that our derivations are correct as they stand, as this is no more than standard calculus associated with working with different kinds of coordinates. However, the reviewer's comments suggest that your presentation of the derivation could be made more transparent. To this end, we have spelled out the different steps of the derivations in more details, which we hope will help the reader reproducing the derivations.

**Regarding Speer's Keff, is (20) correct? It would say Keff = K if K is constant... surely that's not right.**

First note that in (20) K is a tensor so that

$$K\nabla\gamma.n = |\nabla\gamma| \underbrace{(K_i \sin^2(\nabla\gamma, d) + K_d \cos^2(\nabla\gamma, d))}_{=\kappa} \tag{1}$$

$\kappa$ can be a constant only if the mixing is defined as being along the direction of $\nabla\gamma$ (which is not the case here, plus the mixing coefficient would need to be spatially constant). Now let's assume that it is the case,

for instance in a turbulent flow with only molecular mixing. Then we would indeed have $K_{eff}^{speer} = \kappa$ with a total diffusive flux across the $\gamma$ surface equal to $\kappa \int_{z_r=const} \nabla \gamma \cdot \mathbf{ndS}$. However we believe that formula 20 ( which is formula 12 from Speer 1997) is meant to be applied on a climatological (time averaged fields) quantities and not on fully turbulent flows for which we agree that it would be completely useless. In our article we compared our formulation to Speer's formulation in case the reader is more familiar with the latter however it is not used in the remainder of the study and thus not discussed further. We will remove the term 'effective diffusivity' when citing Speer 1997's definition and rather refer to 'average cross isopycnal mixing' which is hopefully less confusing.

**Conversion to density (or any other tracer) coordinates has been done over and over in the literature. Admittedly, it's a messy literature, but it doesn't need to be repeated here. In particular, there is only one effective diffusivity. I haven't read Speer 97, but I've read WD96, Nakamura 96 and Shuckburgh and Haynes 03 (SH03) in great detail — the latter is the most useful. There is an error in the effective diffusivity formulation of WD96 that stems from their incorrect eq. (9), and it seems to have made its way to this paper in (18), though I am not certain because I'm a little uncertain about $\nabla z_r$. I would in any case recommend the authors look at SH03, equations (6) & (7) (and compare to WD96 eq. (12)). That said, I doubt it will affect the results much.**

**Response:** Thank you for the reference to SH03 that we were not aware of, and which we added to the manuscript. However, we disagree that there is only one effective diffusivity. In WD96 or Nakamura 96 for instance, the effective diffusivity is related to the net diatracer flux integrated over an isotracer area divided by a reference laminar area, and scales as $K_{\text{eff}} = \kappa (A_{turbulent}/A_{laminar})^2$ where $\kappa$ is the molecular diffusivity, and $A_{turbulent}/A_{laminar}$ is the ratio of the turbulent isotracer area over a notional laminar value. In that case, the approach is essentially equivalent to the Osborn-Cox (1972) approach to defining a turbulent diffusivity. In our paper, the effective diffusivity however refers to the (turbulent) diffusivity experienced by some material density variable $\gamma$ given a rotated neutral diffusion tensor with isoneutral and dianeutral turbulent mixing coefficients unrelated to the mixing properties of $\gamma$. Moreover, each definition of effective diffusivity tends to have their own specificity and twists, see for instance that of Lee and Nurser, which is not directly related to that of WD96 or Nakamura 96. We feel that our approach is sufficiently context-specific as to warrant going into some details about how we define and compute our effective diffusivities, as we feel that not doing it would be assuming too much from the reader. We were indeed aware of the error in WD96, but a priori, our derivations do not follow that of WD96 and therefore believe that they are correct.